

**Investigation of CATS aerosol products and application toward global diurnal variation of**

**aerosols**

Logan Lee[1], Jianglong Zhang[1], Jeffrey S. Reid[2], and John E. Yorks[3]

[1]Department of Atmospheric Sciences, University of North Dakota, Grand Forks, ND

[2]Marine Meteorology Division, Naval Research Laboratory, Monterey, CA

[3]NASA Goddard Space Flight Center, Greenbelt, MD

Submitted to

ACP

Dec. 2018

Corresponding Author: jzhang@atmos.und.edu; logan.p.lee@und.edu

**Abstract**

We present a comparison of 1064 nm aerosol optical depth (AOD) and aerosol extinction profiles from the Cloud-Aerosol Transport System (CATS) Level 2 aerosol product with collocated Aerosol Robotic Network (AERONET) AOD, Aqua and Terra Moderate Imaging Spectroradiometer (MODIS) Dark Target AOD and Cloud-Aerosol Lidar with Orthogonal Polarization (CALIOP) AOD and extinction data for the period of Mar. 2015-Oct. 2017. Upon quality assurance checks of CATS data, reasonable agreement is found between aerosol data from CATS and other sensors. Using quality assured CATS aerosol data, for the first time, variations in AODs and aerosol extinction profiles are evaluated at 00, 06, 12, and 18 UTC (and/or 0:00 am, 6:00 am, 12:00 pm and 6:00 pm local solar times) on both regional and global scales. This study suggests that marginal variations are found in AOD from a global mean perspective, with the minimum aerosol extinction values found at 6:00 pm (local time) near the surface layer for global oceans, for both the June-November and December-May seasons. Over land, below 500m, the daily minimum and maximum aerosol extinction values are found at 12:00pm and 00:00/06:00 am (local time), respectively. Strong diurnal variations are also found over North Africa and India for the December-May season, and over North Africa, South Africa, Middle East, and India for the June-November season.

## 1.0 Introduction

Aerosol measurement through the sun-synchronous orbits of Terra and Aqua by nature encourages a larger scale, daily average point of view. Yet, we know that pollution (e.g., Zhao et al., 2009; Tiwari et al., 2013; Kaku et al., 2018), fires and smoke properties (e.g., Reid et al., 1999; Giglio et al., 2003; Hyer et al., 2013), and dust (e.g., Mbourou, et al., 1997; Fiedler et al., 2013; Heinold et al., 2013) can exhibit strong diurnal behavior. Sun-synchronous passive satellite aerosol observations from the solar spectrum only provide a small sampling of the full diurnal cycle. Geostationary sensors such as the Advanced Himawari Imager (AHI) on Himawari 8 (Yoshida et al., 2018) and Advanced baseline Imager on GOES-16/17 (Aerosol Product Application Team of the AWG Aerosols/Air Quality/Atmospheric Chemistry Team, 2012) satellites, while an improvement over their predecessors, must overcome the broader range of scattering and zenith angles (Wang et al., 2003; Christopher and Zhang, 2002) with no nighttime retrievals. AErosol RObotic NETwork (AERONET; Holben et al., 1998) based sun photometer studies improve sampling, but until very recently with the development of a prototype lunar photometry mode, are also limited to daylight hours. The critical early morning and evening are largely missed in solar observation-based approaches.

Observations of the diurnal variations of aerosol properties are needed for improving chemical transport modeling, geochemical cycles and ultimately climate. The measurement of diurnal variations of aerosol properties resolved in the vertical is especially crucial for visibility and particulate matter forecasts. Indeed, the periods around sunrise and sunset show significant near surface variability that is difficult to detect with passive sensors. While lidar data from Cloud-Aerosol Lidar with Orthogonal Polarization (CALIOP) provide early afternoon and morning

observations, two temporal points and a 16-day repeat cycle are insufficient to evaluate the critical
morning and evening hours where many key aerosol lifecycle processes take place.
Some of the limiting factors in previous studies can be addressed by the Cloud-Aerosol
Transport System (CATS) lidar that flew aboard the International Space Station (ISS) from 2015
to 2017 (McGill et al. 2015). The ISS's precessing orbit with a $51.6^{\circ}$ inclination allows for 24
hour sampling of the tropics to mid-latitudes, with the ability to observe aerosol and cloud vertical
distributions at both day and night time with high temporal resolution. For a given location within
$\pm 51.6^{\circ}$ (Latitude), after aggregating roughly 60 days of data, near full diurnal cycle of aerosol and
cloud properties can be obtained from CATS observations (Yorks et al. 2016). This provides a
new opportunity for studying diurnal variations (day and night) in aerosol vertical distributions
from space observations.
Use of CATS has its own challenges. Most importantly, CATS retrievals must cope with
variable solar noise around the solar terminator where we expect some of the strongest diurnal
variability to exist. Further, CATS lost its 532 nm channel early in its deployment, leaving only a
1064 nm channel functioning. The availability of only one wavelength limited the CATS cloud-
aerosol discrimination algorithm, which can cause a loss of accuracy compared to CALIPSO
which has 2 wavelengths. This deficiency is in part overcome by using the Feature Type Score
(CATS Algorithm Theoretical Basis Document). Using two years of observations from CATS, in
this paper, we focus on understanding of the following questions: How well do CATS derived
aerosol optical depth (AOD) and aerosol vertical distributions compare with aerosol properties
derived from other ground-based and satellite observations such as AERONET, MODIS and
CALIOP? Do differences exhibit a diurnal cycle? What are the diurnal variations of aerosol optical
depth on a global domain?  What are the diurnal variations of aerosol vertical distribution on both
regional and global scales?

**2.0     Datasets**
Four datasets, including ground-based AERONET data, as well as satellite retrieved
aerosol properties from MODIS and CALIOP, are used for inter-comparing with AOD and aerosol
vertical distributions from CATS.  Upon thorough evaluation and quality assurance procedures,
CATS data are further used for studying diurnal variations of AOD and aerosol vertical
distributions for the period of Mar. 2015 – Oct. 2017.

**2.1 CATS**
CATS Level 2 (L2) Version 3-00 5 km Aerosol Profile products (L2O_D-M7.2-V3-
00_05kmPro, L2O_N-M7.2-V3-00_05kmPro) were used in this study for nearly the entire period
of CATS operation on the ISS (~Mar. 2015–Oct. 2017).  CATS L2 profile data is provided at 5
km along-track horizontal resolution and 533 vertical levels at 60 m vertical resolution and a
wavelength of 1064 nm.  CATS also provides data at 532 nm, but due to a laser-stabilization issue,
532 nm data is not recommended for use (Yorks et al. 2016).  Thus, only 1064 nm products were
used in this study.  Although the uncertainties in CATS aerosol retrievals have not yet been
documented for the CATS V3-00 extinction and AOD products, much like CALIOP, uncertainties
in the calibration and assumed lidar ratios are the primary contributors to the extinction and AOD
uncertainties. The uncertainties in the CATS 1064 nm attenuated total backscatter (ATB) is on the
order of 7-10% for nighttime and is around 20% for daytime (Pauly et al., 2019), while the
uncertainties in the assumed 1064 nm lidar ratios for CATS are 30%. Thus, the CATS 1064 nm
extinction (40-70%) and AOD (30-50%) uncertainties are very similar to the corresponding
CALIOP 1064 nm uncertainties.

CATS data are quality-assured following a manner similar to Campbell et al. (2012), which

was applied to CALIOP.  QA thresholds (including extinction QC flag, Feature Type Score, and
uncertainty in extinction coefficient) are listed below:

(a) Extinction_QC_Flag_1064_Fore_FOV is equal to 0 (non-opaque layer; lidar ratio

unchanged)

(b) Feature_Type_Fore_FOV = 3 (contains aerosols only)

(c) -10 <= Feature_Type_Score_FOV <= -2 (Feature Type Score < 0 is aerosol, with -10

being complete confidence, and 0 being as likely to be cloud as aerosol)

(d) Extinction_Coefficient_Uncertainty_1064_Fore_FOV $<= 10 \; km^{-1}$

Extinction was also constrained using a threshold as provided in the CATS data catalog

(Extincton_Coefficient_1064_Fore_FOV <= 1.25 km$^{-1}$), similar to several previous studies
(Redemann et al., 2012; Toth et al., 2016).  Only profiles with extinction coefficient values less
than 1.25 km$^{-1}$ are included in this study.  Small negative extinction coefficient values, however,
are included in aerosol profile related analysis, to reduce potential high biases in computed mean
profiles.  Note that a similar approach has also been conducted in deriving passive-based AOD
climatology (e.g. Remer et al., 2005). For this study, both the
Aerosol_Optical_Depth_1064_Fore_FOV and Extinction_Coefficient_1064_Fore_FOV datasets
were used to provide AOD and 1064 nm extinction profiles (hereafter the term "extinction" will
refer to 1064 nm unless explicitly stated otherwise), respectively.

**2.2 CALIOP**
NASA's CALIOP is an elastic backscatter lidar that operates at both 532 nm and 1064 nm
wavelengths (Winker et al., 2009).  Being a part of the A-Train constellation (Stephens et al.,
2002), CALIOP provides both day- and night-time observations of Earth's atmospheric system, at
a sun-synchronous orbit, with a laser spot size of around 70 m and a temporal resolution of ~16
days (Winker et al., 2009).  For this study, CALIOP Level 2.0 Version 4.1 5 km Aerosol Profile
products (L2_05kmAProf) are used for inter-comparing to CATS retrieved AODs and aerosol
vertical distributions.
L2_05kmAProf data are available at 5 km horizontal resolution along-track and include
aerosol retrievals at both 532 nm and 1064 nm wavelengths.  The vertical resolution is 60 m near-
surface, degrading to 180 m above 20.2 km in MSL altitude.  As only 1064 nm CATS data are
used in this study as mentioned above, likewise only those CALIOP parameters relating to 1064
nm are used in this study (Vaughan et al., 2019; Omar et al., 2013). Note that as suggested by
Rajapakshe et al. (2017), lower signal-to-noise ratio (SNR) and higher minimum detectable
backscatter are found for the CALIOP 1064 nm data in-comparing with the CALIOP 532 nm data.
Also, the CALIOP aerosol layers are detected at 532 nm and the 1064 nm extinction is only
computed for the bins within these layers.  This may introduce a bias for aerosol above cloud
studies.  The uncertainties in retrieved aerosol extinction, as suggested by Young et al., (2013), is
around 0.05–0.5 km$^{-1}$ for the 532 nm channel.  Validated against AERONET data, Omar et al.,
(2013) suggested that 74% and 81% of the CALIOP AOD retrievals fall within the expected
uncertainties (0.05+0.4*AOD) as suggested by Winker et al., (2009) for the 1064nm channel, for
all sky and clear sky conditions respectively.
In        this        study,        Extinction_Coefficient_1064        and
Column_Optical_Depth_Tropospheric_Aerosols_1064 are used for CALIOP extinction and AOD
retrievals, respectively (Vaughan et al., 2019; Omar et al., 2013).  As with the CATS data, CALIOP
data are quality-assured following the quality assurance steps as mentioned in a few previous
studies (e.g. Campbell et al., 2012; Toth et al., 2016; 2018).  These QA thresholds are listed below:
(a)  Extinction_QC_Flag_1064 is equal to 0 (unconstrained retrieval; initial lidar ratio
unchanged)
(b) Atmospheric_Volume_Description = 3 or 4 (contains aerosols only)
(c) $-100 <= CAD\_Score <= -20$  (CAD < 0 is aerosol, with -100 being complete confidence,
and 0 being as likely to be cloud as aerosol)
(d) Extinction_Coefficient_Uncertainty_1064 $<= 10$  $km^{-1}$
Furthermore, as in Campbell et al. (2012), only those profiles with AOD > 0 were retained
in order to avoid profiles composed of only retrieval fill values.  Extinction was also constrained
to the nominal range provided in the CALIOP data catalog (Extinction_1064 <= 1.25 km[-1]), similar
to our QA procedure for CATS as described above.

**2.3 MODIS Collection 6.1 Dark Target product**
Moderate Resolution Imaging Spectroradiometer (MODIS) Aqua and Terra Collection 6.1
Dark Target over-ocean AOD data (Levy et al., 2013) were used for comparison to CATS AOD.
The data field of "Effective_Optical_Depth_Best_Ocean" was used and only those data flagged as
"good" or "very good" by the Quality_Assurance_Ocean runtime QA flags were selected for this
study, similar to Toth et al. (2018).  Because MODIS does not provide AOD in the 1064 nm
wavelength, AOD retrievals from 860 and 1240 nm spectral channels are used to logarithmically
interpolate AODs at 1064 nm.  Here we assume the Ångström Exponent  value, computed using
instantaneous AOD retrievals at the 860 and 1240 nm, remains the same for the 860 to 1064 nm
wavelength range, similar to what has been suggested by Shi et al., (2011; 2013). Mean and
standard deviation of Ångström exponents using this method were 0.69 and 0.55, respectively.
Only totally cloud free (or cloud fraction equal to zero) retrievals, as indicated by the
Cloud_Fraction_Land_Ocean parameter, are used. While the uncertainties in MODIS infrared
(e.g. 1240 nm) retrievals are less explored, the reported over ocean MODIS DT AOD retrievals
are ($+(0.04 + 0.1*$AOD$, -(0.02 + 0.1*$AOD$))$ for the green channel (Levy et al., 2013).

**2.4 AERONET**

By measuring direct and diffuse solar energy, AERONET observations are used for
retrieving AOD and other ancillary aerosol properties such as size distributions (Holben et al.,
1998). AERONET data are considered as the ground truth for evaluating CATS retrievals in this
study. Only cloud screened and quality assured version 3 level 2 AERONET data at the 1020 nm
spectrum are selected and are used for inter-comparing with CATS AOD retrievals at the 1064 nm
wavelength. AERONET does not have specific guidance on error in the 1020 nm channel, as it
is known to have some thermal sensitivities. However, they do report significantly more
confidence in version 3 of the data, which has temperature correction (Giles et al., 2019). Error
models are ongoing, and for this study we assume double the RMSE, or +/-0.03. Note that Version
3 AERONET data are designed to reduce thin cirrus cloud contamination as well as rescue heave
aerosol scenes that were misclassified as clouds in previous versions (e.g. Giles et al., 2019).

**3.0    Results & Discussion**

**3.1 Inter-comparison of CATS data with AERONET, MODIS and CALIOP data**

Note that most evaluation efforts for passive and active sensor AOD retrievals are focused
on the visible spectrum and the performance of AOD retrievals at the 1064 nm channel is less
explored.  Thus, in this sub-section, the performance of over land and over ocean CATS AOD
retrievals are compared against AERONET and C6.1 over ocean MODIS Dark Target (DT)
aerosol products.  In AOD related studies, CATS and CALIOP reported AOD values are used.
However, only AOD values with corresponding aerosol vertical extinction that meet the QA
criteria as mentioned in Sections 2.1 and 2.2 were used. CATS derived aerosol extinction vertical
distributions are also cross-compared against collocated CALIOP aerosol extinction vertical
distributions.

**3.1.1 CATS-AERONET**
As the initial check, CATS data from nearly the entire mission (Mar. 2015-Oct. 2017) were
spatially (within 0.4 degree Latitude and Longitude) and temporally (±30minutes) collocated
against ground-based AERONET data.  Note that one AERONET measurement may be associated
with several CATS retrievals in both space and time, and vice versa. Thus, both CATS and
AERONET data were further averaged spatially and temporally, which results in only one pair of
collocated and averaged CATS and AERONET data for a given collocated incident.  Also, only
data pairs with AOD larger than 0 from both instruments are used for the analysis.  This step is
necessary to exclude CATS profiles with all retrieval fill values as discussed in Section 2 (Toth et
al., 2018).  Such profiles containing all retrieval fill values were found to make up approximately
5.3% of all CATS profiles in the dataset.   Note that the CATS-AERONET comparisons are for
daytime only, and higher uncertainties are expected for CATS daytime than nighttime AODs.
As shown in Figure 1a, without quality-assurance procedures, high spikes in CATS AOD
of above 1 (1064 nm) can be found for collocated AERONET data with AOD less than 0.4 (1020
nm).  Still, those high spikes in CATS AOD are much reduced compared to the V2-01 CATS
aerosol products  (e.g. a similar plot as Figure 1 is included in the Appendix A with the use of V2-
01 CATS aerosol data).  Upon completion of the QA steps as outlined in Section 2.1, a reasonable
agreement is found between quality-assured CATS (1064 nm) vs. AERONET (1020 nm) AODs
with a correlation of 0.65 (Figure 1b).  Comparing Figure 1a with 1b, with the loss of only ~1-2%
of collocated pairs due to the QA procedures, we have observed an overall improvement in
correlation between CATS and AERONET AOD from 0.51 to 0.64, thus, only QAed CATS data
are used hereafter.  We also found that requiring the Extinction QC flag to be equal to 0 and the
Extinction Uncertainty to be less than 10 $km^{-1}$ had the largest impacts on reducing the difference
in mean and medians of the AERONET and CATS AOD.  Still, this exercise highlights the need
for careful quality checks of the CATS data before applying the CATS data for advanced
applications to overcome cloud-aerosol discrimination uncertainties.

**3.1.2 CATS-MODIS**
To examine over ocean performance, column integrated CATS AODs are inter-compared
with collocated Terra and Aqua C6.1 MODIS DT over ocean AOD, interpolated to 1064 nm.  Over
ocean C6.1 MODIS DT data are selected due to the fact that higher accuracies are reported for
over ocean versus over land MODIS DT AOD retrievals (Levy et al., 2013). In addition, compared
to over land MODIS DT data, which provide AOD retrievals at three discrete wavelengths (0.46,
0.55 and 0.65 µm), over water AOD retrievals are available from 7 wavelengths including the 0.87
and 1.24 µm spectral channels, allowing a comparison with CATS AOD at the same wavelength
upon logarithmic interpolation, again, assuming the aerosol Ångström Exponent value remains
unchanged from 0.87 to 1.064 µm as well as from the 1.064 µm to 1.24 µm spectral channels.

MODIS and CATS AOT retrievals are collocated for the study period of Mar. 2015-Oct.

2017 (Figure 2).  Pairs of CATS and MODIS data were first selected for both retrievals that fall
within ±30 minutes and 0.4 degrees latitude and longitude of each other.  Then, similar to the
AERONET and CATS collocation procedures, collocated pairs were further averaged to construct
one pair of collocated MODIS and CATS data for a given collocation incident.  Shown in Figure
2a, a correlation of 0.72 is found between collocated over water Terra MODIS C6.1 DT and CATS
AODs with a slope of 0.74.  Similar results are found for the comparisons between over water
Aqua MODIS and CATS AODs with a correlation of 0.74 and a slope of 0.70.

**3.1.3 CATS-CALIOP AOD**

In the previous two sections, AODs from CATS were inter-compared with retrievals from

passive-based sensors such as MODIS and AERONET.  In this section, AOD data from CALIOP,
which is an active sensor, are evaluated against AOD retrievals from CATS.  Note that despite
difference in instrumental designs, CALIOP and CATS are both elastic backscatter lidars.  Again,
for each collocation incident, pairs of CALIOP and CATS data are selected in which both retrievals
fall within ±30 minutes temporally and 0.4 degrees latitude and longitude spatially.  There could
be multiple CATS retrievals corresponding to one CALIOP data point, and vice versa.  Thus, the
collocated pairs are further averaged in such a way that only one pair of collocated CATS and
CALIOP data is derived for each collocation incident.  .

Figure 3a shows the comparison of CATS and CALIOP AODs for all collocated pairs

including both day- and night-time.  A reasonable correlation of 0.74, with a slope of 0.73, is found
for a total of 2762 collocated data pairs.  Further breaking down the comparison into day and night
cases, a much better agreement is found between the two datasets during nighttime with
correlations of 0.81 and 0.83 for over-ocean and over-land cases respectively.  In comparison, a
lower correlation of 0.64, with a slope of 0.49, is found between the two datasets, using over land
daytime data only, for a total of 170 collocated pairs.  Correspondingly, a lower correlation of
0.55, with a slope of 0.57, is found between the two datasets, using over ocean daytime data only,
for a total of 1180 collocated pairs.  This result is not surprising as daytime data from both CALIOP
and CATS are nosier due to solar contamination (e.g. Omar et al., 2013; Toth et al., 2016).

Note that based on the slopes of the regression lines shown in Figures 1-3, AODs retrieved

by CATS are less than AERONET, CALIOP and DT Aqua MODIS AOD retrievals.   As shown
in Table 1, however, for the one-to-one collocated datasets, mean CATS AODs (1064 nm) are
~10% higher than AERONET AODs (1020 nm).   The CATS AODs are ~3% higher than CALIOP
AOD (1064 nm) and are ~5-10% higher than DT MODIS AODs.  One possible explanation for
this discrepancy is because mean AODs are dominated by low AOD cases and the slopes of the
regression relationships are strongly affected by a few high AOD cases.   Thus, it is likely that
CATS AODs are overestimated at the low AOD ranges and are underestimated at the high AOD
ranges.

Also note that as suggested by Omar et al., (2013), the choices of spatial and temporal

collocation windows have an effect on collocation results.  Thus, we repeated the exercises in Figures
1-3 by doubling the spatial and temporal collocation windows as well as reducing the collocation
windows by half.  The descriptive statistics of this sensitivity study are included in Table 2.  While
the number of collocated data pairs are drastically affected by the spatial and temporal collocation
window sizes, less significant changes are found in descriptive statistics such as mean, median,
and standard deviations of AODs, as well as slopes and correlation values. The slope of DT Aqua
MODIS and CATS AODs, however, seems sensitive to changes in collocation methods. Changes
in slope of 0.61 to 0.78 are found for the change of temporal collocation window from 15 minutes
to 60 minutes with a fixed spatial collocation window of 0.4° Latitude/Longitude.

Still, larger discrepancies between CATS and CALIOP AODs during daytime indicate that

both sensors are susceptible to solar contamination. To overcome solar contamination and more
accurately detect aerosol layers, CALIOP and CATS data products are averaged up to 80 km and
60 km, respectively. Noel et al. (2018) found that the feature type score can be used for cloud
screening throughout the diurnal envelope of solar angles. To further evaluate impact of the solar
contamination introduced bias in the diurnal analysis in aerosol detection or products, CATS
AODs are evaluated as a function of local time. For each CATS observation of a given location
and UTC time, the associated local time is computed by adding the UTC time by 1 hour per 15°
Longitude away from the Prime Meridian in the east direction. Figure 4a shows the CATS AOD
versus local time for both global land and oceans, constructed using 6 hourly mean CATS AOD
binned on a 5 degree by 5 degree grid globally. While the data has additional noise, no major
deviations in AODs are found during either sunrise or sunset time, although we speculate that
larger uncertainties in CATS AODs and extinctions may be present around day and night
terminators. Figure 4b shows a similar plot as Figure 4a, but with the region restricted to 25°S-
52°S. Here, we want to investigate the variations in CATS AODs as a function of local time, over
relatively aerosol free oceans. We picked 25°S as the cutoff line as CATS data only available to
51.6°S (limited to the ISS inclination angle) and thus, this threshold is used to ensure enough data
samples in the analysis, although some land regions are also included. As indicated in Figure 4b,
again, no significant deviations in pattern are found for both sunrise and sunset time, plausibly
indicating that solar contamination, as speculated, may not be as significant. Comparing the mean
AOD at local midnight to the mean AOD at local noon by performing a student's t test, the
difference is not significant at the 95% confidence level, with a p-value of 0.16.

Figure 4c shows the difference between AERONET (1020 nm) and CATS (1064 nm) AOD

(ΔAOD) as a function of local time. Again, although data are rather noisy, no major pattern is
found near sunrise or sunset times, further indicating that solar contamination during dawn or dusk
times may have a less severe impact to CATS AOD retrievals from a long term mean perspective.
In summary, Sections 3.1.1-3.1.3 suggest that with careful QA procedures, AOD retrievals from
CATS are comparable to those from other existing sensors such as AERONET, MODIS, and
CALIOP at the same local times.

**3.1.4 CATS-CALIOP Vertical Extinction Profiles**

One advantage of CATS is its ability to retrieve both column-integrated AOD and vertical

distributions of aerosol extinction. Therefore, in this section, extinction profiles from CATS are
compared with that from CALIOP. Again, similar to the Section 3.1.3, collocated profiles for
CATS and CALIOP are first found for both retrievals that are close in space and time (within ±30
minutes and 0.4 degrees latitude and longitude). However, different from Section 3.1.3, only one
pair of collocated CATS and CALIOP profiles, which has the closest Euclidian distance on the
earth's surface, is retained for each collocated incident.

The CATS cloud-aerosol discrimination (CAD) algorithm is a multidimensional

probability density function (PDF) technique that is based on the CALIPSO algorithm (Liu et al.
2009). The PDFs were developed based on Cloud Physics Lidar (CPL) measurements obtained
during over 11 field campaigns and 10 years. As shown in Figure 5e, a reasonable agreement is
found between CATS V3-00 aerosol extinction with CALIOP for over land. However, CATS
overestimates aerosol extinction around 1 km compared to CALIOP over ocean (Figure 5d). This
can also be seen on a plot of the difference between CATS and CALIOP 1064 nm extinction for
all collocated profiles, included in Figure 5f, where there is an overall positive difference around
1 km.

Due to the precessing orbit of the ISS, the CATS sampling is irregular and very different

compared to the sun-synchronous orbits of the A-Train sensors. These orbital differences between
CATS and CALIOP make comparing the data from these two sensors challenging since they are
fundamentally observing different locations of the Earth at different times. Thus, we shouldn't
expect the extinction profiles and AOD from these two sensors to completely agree. Additionally,
there are other algorithm and instrument differences that can lead to differences in extinction
coefficients and AOD. Over land where dust is the dominant aerosol type, differences in lidar
ratios between the two retrieval algorithms (CATS uses 40 sr while CALIOP uses 44 sr), can cause
CATS extinction coefficients that are up to 10% lower than CALIOP, potentially explaining the
higher CALIOP extinction values in Figure 5e. Over ocean, especially during daytime, differences
in CATS and CALIOP lidar ratios for marine and smoke aerosols can introduce a difference
between CATS and CALIOP extinction coefficients (Figure 5d). These difference in over ocean
data (Figure 5d) could also attributed to differences in CATS and CALIOP 1064 nm backscatter
calibration.  For example, Pauly et al. (2019) reports that CATS attenuated total backscatter is
about 19.7% lower than PollyXT measurements in the free troposphere and 19% lower than
CALIOP opaque cirrus clouds due to calibration uncertainties for both sensors.

Also, differences in the lowest 250 m between CATS and CALIOP extinction profiles are

observable, which are due to how the instrument algorithms detect the surface and near-surface
aerosols. Both the CATS and CALIOP feature detection algorithms create a gap between the
surface and near-surface aerosol base altitude, despite the possible presence of aerosols in this
altitude region. CALIOP has an aerosol base extension algorithm that is designed to (1) detect
scenarios when aerosols are present in the bins just above the surface and (2) extend the near-
surface aerosol layer base down to the surface (Tackett et al., 2018). However, CATS does not use
such an algorithm so false regions of "clear-air" exist between the surface and near-surface aerosol
layers.
Vertical profiles of collocated CATS and CALIOP extinction for daytime only profiles and
nighttime only profiles are shown in Figure 5b and 5c, respectively. Compared to a total collocated
pair count of 2748 in the overall profile data, day and night profiles have 1311 and 1437 collocated
pairs, respectively. Again, the shapes of the CATS and the CALIOP nm extinction vertical profile
are similar for all three cases, despite the above mentioned offsets in altitude. Figure 5d and 5e
show the mean of those extinction profiles which occurred over-water and over-land, as defined
by the CATS surface type flag. Again in both cases CATS and CALIOP have similar shapes in
their vertical extinction profiles. The vertical structure of over-water extinction is also very similar
to that of all profiles, day, and night, which is perhaps not surprising as water profiles made up
2142 of 2748 (~78%) collocated pairs. The vertical structure of over-land is more different than
the other groups, as the extinction is higher throughout a larger depth of the atmosphere, tapering
off much more slowly from the surface. Furthermore, the extinction from CATS is actually lower
than CALIOP for over-land profiles, unlike all other categories.

**3.2 Diurnal Cycle of AODs and Aerosol Vertical Distributions**
Using the QAed CATS data, seasonal variations as well as diurnal variations in CATS
AODs are derived in this section. Diurnal variations in the vertical distributions of CATS aerosol
extinction are also examined at both global and regional scales.

**3.2.1 Seasonal and Diurnal Variation of AOD**
Figures 6a-b show the spatial distributions of CATS AODs at the 1064 nm spectral channel
for boreal winter-spring (Dec.-May, DJFMAM) and boreal summer-fall (June-Nov, JJASON)
seasons, for the period of Mar. 2015-Oct. 2017.  To construct Figures 6a and 6b, quality-assured
CATS AODs are first binned on a 5 degree by 5 degree grid over the globe for the above mentioned
two bi-seasons.  For each $5\times5°$ (Latitude/Longitude) bin, for a given season, CATS AODs are
averaged on a pass-basis first, and then further averaged seasonally to represent AOD value of the
given bin.  Both daytime and nighttime retrievals are included in this Figure, as well as Figures 7-

9.

In DJFMAM season, significant aerosol features are found over North Africa, Middle East,
India and Eastern China.  For the JJASON season, besides the above mentioned regions, aerosol
plumes are also observable over Southern Africa, related to summer biomass burning of the region
(e.g. Eck et al., 2013). The seasonal-based spatial distributions of AODs from CATS, although
reported at the 1064 nm channel which is different from the 550 nm channel that is conventionally
used, are similar to some published results (e.g. Lynch et al., 2016).
For comparison purposes, Figures 6c-6d shows similar plots to Figures 6a-6b, but with the
use of CALIOP AOD at the 1064 nm spectral channel.  Note that those are climatological means
rather than pairwise comparisons.  While patterns are similar in general, at regions with peak
AODs of 0.4 or above for CALIOP, such as North Africa for the DJFMAM season and North
Africa, Middle East and India for the JJASON, much lower AODs are found for CATS. In some
other regions, such as over South Africa for the JJASON season, however, higher CATS AOD
values are observed. A table of mean AOD across each of these regions as well as over the globe
(within the latitude range where CATS has data) has been included for reference (Tables 3).
Figures 6e and 6f show the similar spatial plots as Figures 6a and 6b but with the use of Aqua
MODIS AODs from the DT products (using all available MODIS DT retrievals that passed QA
steps as described in Section 2.3). For the Aqua MODIS DT products, aerosol retrievals at the
short-wave Infra-red channels are only available over oceans, and thus Figures 6e-6f show only
over ocean retrievals. Again, while general AOD patterns look similar, discrepancies are also
visible, such as over the coast of south west Africa for the JJASON season and over the west coast
of Africa for the DJFMAM season. Those discrepancies may result from biases in each product,
but it is also possibly due to the differences in satellite overpass times, as CALIOP provides early
morning and afternoon over passes, and Aqua MODIS has an over pass time after local noon,
while CATS is able to report atmospheric aerosol distributions at multiple times during a day.

Similar to Figures 6a and 6b, Figures 7a and 7b show the spatial distribution of CATS

AODs, but for CATS extinction values that are below 1 km Above Ground Level (AGL) only, for
the DJFMAM and JJASON seasons respectively. Figure 7c and 7d show the CATS mean AOD
plots for extinction values from 1-2 km AGL, while Figure 7e and 7f show CATS mean AOD for
extinction values above 2 km AGL. For the DJFMAM season, elevated aerosol plumes with
altitude above 2 km AGL are found over the North Africa. For the JJASON season, elevated dust
plumes (> 2 km AGL) are found over North Africa and the Middle East regions, while elevated
smoke plumes are found over the west coast of South Africa where above cloud smoke plumes are
often observed during the Northern hemispheric summer season (e.g. Alfaro-Contreras et al.,

2016).

CATS has a non-sun-synchronized orbit, which enables measurements at nearly all solar
angles.  Thus, we also constructed 5×5° (Latitude/Longitude) gridded seasonal averages (for
DJFMAM and JJASON seasons) of CATS AODs at 0, 6, 12 and 18 UTC that represent 4 distinct
times in a full diurnal cycle, as shown in Figure 8.  To construct the seasonal averages, observations
within ±3 hours of a given UTC time as mentioned above are averaged to represent AODs for the
given UTC time.  On a global average, the mean AODs are 0.090, 0.089, 0.088 and 0.089 for 0, 6,
12 and 18 UTC respectively for the JJASON season and are 0.099, 0.096, 0.093 and 0.093 for the
DJFMAM season.  Thus, no significant diurnal variations are found on a global scale,.
Still, strong diurnal variations with the maximum averaged diurnal AOD changes of above
0.10 can be observed for regions with significant aerosol events such as Northern Africa, Middle
East and India for the DJFMAM season and Northern Africa, Southern Africa, Middle East and
India for the JJASON season, as illustrated in Figure 9.  Note that Fig. 9a shows the maximum
minus minimum seasonal mean AODs for the four difference times as shown in Figs. 8a,c,e,g.
Similarly, Fig. 9b shows the maximum minus minimum seasonal mean AODs for the four
difference times as shown in Figs. 8b,d,f,h. Interestingly but not unexpectedly, regions with
maximum diurnal variations match well with locations of heavy aerosol plumes as shown in
Figures 6 and 8.

**3.2.2 Diurnal variations of Aerosol Extinction on a Global Scale (both at UTC and local time)**
Using quality-assured CATS derived aerosol vertical distributions, mean global CATS
extinction vertical profiles are also generated as shown in Figure 10.  Similar to steps as described
in the section 3.2.1, CATS extinction profiles are binned into 00, 06, 12, and 18 UTC times based
on the closest match in time for the JJASON and DJFMAM seasons.  Figure 10a shows the daily
averaged CATS extinction profiles in a black line, and 00, 06, 12 and 18 UTC averaged in blue,
green, yellow and red lines respectively, for the DJFMAM season. Similar plot is shown in Figure
10d for the JJASON season.  CATS extinction profiles for the daily average as well averages for
the four selected times are similar, suggesting that minor temporal variations in CATS extinctions
can be expected for global averages.

Those global averages are dominated by CATS profiles from global oceans (Figure 10b

and 10e), which also have small diurnal variations, as ~70% of the globe is covered by water.  In
comparison, noticeable diurnal changes in aerosol vertical distributions are found over land as
shown in Figure 10c and 10f.  For the DJFMAM season, at the 1 km altitude, the minimum and
maximum aerosol extinctions are at 12 and 18 UTC respectively.  Similarly, the minimum and
maximum aerosol extinctions are at 12 and 00 UTC at the altitude of 400 m.  For the JJASON
season, the minimum aerosol extinction values are found at 12 UTC for the whole 0-2 km column,
while the maximum aerosol extinction values are at 18UTC for 1.5 km and 00 UTC for the 300-
400 m altitude.  Still, it should be noted that aerosol concentrations may be a function of local
time, yet for a given UTC time, local times will vary by region.  Also, due to solar contamination,
nighttime retrievals from CATS are significantly and demonstrably less noisy than daytime retrievals,
and this difference in sensor sensitivity between day and night may further affect the derived
diurnal variations in CATS AOD and aerosol vertical profiles as shown in Figure 3 for individual
retrievals.   Still, no apparent solar pattern is detectable from Figure 8, and only minor diurnal
variations are found for Figure 10a and 10d, which indicate that such a solar contamination may
introduce noise but not bias to daytime aerosol retrievals, from a global mean perspective.

If we examine the mean global CATS extinction vertical profiles with respect to local time

as shown in Figure 11, however, some distinct features appear.  For example, Figure 11a and 11d
suggests that on global average, the minimum aerosol extinction below 1 km is found for 6:00 pm
local time, for both JJASON and DJFMAM seasons.  Similar patterns are also observed for over
global oceans.  However, for over land cases, for both seasons, the minimum and maximum aerosol
extinction below 600 m is found for 12:00 pm  and 00:00/06:00 am local time.

### 3.2.3 Diurnal variations of Aerosol Extinction on a Regional Scale (at local time)

In this section, the diurnal variations of aerosol vertical distributions are studied as a

function of local solar time for selected regions with high mean AODs as highlighted in Figure 6.
We picked local solar time here as for those regional analyses. Note a near 1 to 1 transformation
can be achieved between UTC and local solar time. Also, as learned from the previous section,
aerosol features are likely to have a local time dependency. A total of four regions, including
Africa-North, Middle East, India and Northeast China, which show significant seasonal mean
AODs in Figure 6, are selected for the DJFMAM season (Figure 12). For the JJASON season
(Figure 13), in addition to the above mentioned 4 regions, the Africa-South region is also included
due to biomass burning in the region during the Northern Hemisphere summer time.  The
Latitude/Longitude boundary of each selected region is described in Table 4.  Regional-based
analyses are also conducted for 4 selected regions for the DJFMAM season and 5 selected regions
for the JJASON season at four local times: 0:00 am (midnight), 6:00 am, 12:00 pm and 6:00 pm,
using quality assured CATS profiles.  Generally, the maximum diurnal change in aerosol
extinction is found at the altitude of below 1 km for all regions as well for both seasons.  Also,
larger diurnal variations in vertical distributions of aerosol extinction are found for the JJASON
season, in-comparing with the DJFMAM season, while regional-based differences are apparent.
For the Africa-North region, dominant aerosol types are dust and smoke aerosol for the
DJFMAM season, and dust for the JJASON season (e.g. Remer et al., 2008).  Interestingly, the
maximum aerosol extinction below 500m is found at 6:00 am for the DJFMAM season. While for
the JJASON season, the maximum aerosol extinctions are found at 0:00 am / 6:00 am for the 100-
500 m layer, with a significant ~10-20% higher aerosol extinction from the daily mean.  Note that
6:00 am in the Africa-North region corresponds to early morning, which has been identified in
several studies (Fiedler et al., 2013; Ryder et al. 2015) as the time of day when nocturnal low-level
jet breakdown causes large amounts of dust emission in this region.  Thus, we suspect that this
6:00 am peak in maximum aerosol extinctions may be the signal resulting from the low-level jet
ejection mechanism captured on a regional scale.  As the day progresses into the afternoon and
early evening, we find the aerosol heights shifting upwards, likely related to the boundary layer's
mixed layer development.
For the Middle East region, for the JJASON season, a daily maximum in aerosol extinction
of ~0.15 km$^{-1}$ is found at midnight (0:00 am) , with a daily minimum of ~0.08 km$^{-1}$ found at local
noon (12:00 pm), for the peak aerosol extinction layer that has a daily mean aerosol extinction of
~~0.12 km$^{-1}$.  This translates to a ~±20-30% daily variation for aerosol extinction for the peak
aerosol extinction layer. Smaller daily variation in aerosol extinction, however, is found for the
same region for the DJFMAM season.
For the India region, for the JJASON season, a large peak in aerosol extinction of up to
10% higher than daily mean is found at 6:00 am below 500 m.  The minimum aerosol extinction
is found at 12:00/6:00 pm for the layer below 500 m, and is overall ~10% lower than the peak
daily mean aerosol extinction value.  For the DJFMAM season, minimum aerosol extinctions are
found at 12:00 pm for near the whole 0-2 km column, while for the layer below 500 m, the
maximum aerosol extinction values are found at mid-night (0:00 am).

For the Northeast China region , a significant peak found at the 500m-1km layer for local

afternoon (6:00 pm) for the DJFMAM season. A similar feature is also found for the JJASON
season.  While the peak extinction for the JJASON season happens at 06:00am for the aerosol
layer below 500m.   Lastly, for the Africa-South region, biomass burning aerosols are prevalent
during the summer time and thus only the JJASON season is analyzed.  As shown in 13b, below
500m in altitude, lower extinction values are found for local afternoon (18:00 pm) and higher
extinction values are found for local morning or early morning (0:00 and 6:00 am).

**4.0    Conclusions**

Using CALIOP, MODIS and AERONET data, we evaluated CATS derived AODs as well

as vertical distributions of aerosol extinctions for the study period of for Mar. 2015 – Oct. 2017.
CATS data (at 1064 nm) were further used to study variations in AODs and aerosol vertical
distributions diurnally.  We found:

(1) Quality assurance steps are critical for applying CATS data in aerosol related

applications.  With a less than 2% data loss due to QA steps, an improvement in

correlation from 0.51 to 0.65 is found for the collocated CATS and AERONET AOD

comparisons.  Using quality assured CATS data, reasonable agreements are found

between CATS derived AODs and AODs from CALIOP, Aqua MODIS DT and Terra

MODIS DT at the same local times, with correlations of 0.74, 0.74 and 0.72

respectively.

(2) While the averaged vertical distributions from CATS compare reasonably well with

that from CALIOP, differences in peak extinction altitudes are present.  This may due

to sampling difference as well as algorithm and instrument differences such as different

lidar ratios used.

(3) From the global mean perspective, minor changes are found for AODs at four selected

549         times, namely 00, 06, 12 and 18 UTC.  Yet noticeable diurnal variations in AODs of

above 0.10 (at 1064 nm) are found for regions with extensive aerosol events, such as

over North Africa, Middle East, and India for the DJFMAM season, and over North

and South of Africa, India and Middle East for the JJASON season.

(4) From the global mean perspective, changes are less noticeable for the averaged aerosol

extinction profiles at 00, 06, 12 and 18 UTC.  Yet, if the study is repeated with respect

to local time, a peak in aerosol extinction is found for local noon (12:00pm) for the

DJFMAM season and the minimum value in aerosol extinction is found at 6:00 pm

local time for both JJASON and DJFMAM seasons.  While the over water aerosol

vertical distributions are similar to the global means, for over land cases, the minimum

and maximum extinctions are found at local noon (12:00pm) and local morning or early

morning (6:00am and 0:00am) for the layer below 500 m for both seasons.

(5) Larger diurnal variations are found in regions with heavy aerosol plumes such as North

and South (summer season only) of Africa, Middle East, India and Eastern China.  In

particular, aerosol extinctions from 6:00 am over North Africa are ~10% higher than

daily means for the 0-500 m column for both  seasons. We suspect this may be related

to increase in dust concentrations due to breakdown of low level jets at early morning

time for the region.

(6) Still, readers should be aware that AOD retrievals at the 1064 nm are less sensitive to

fine mode aerosols such as smoke and pollutant aerosols, compared to coarse mode

aerosols such as dust aerosols (e.g. Dubovik et al., 2000). Thus, an investigation of

diurnal variations of aerosol properties at the visible channel may be also needed for a

future study.

This paper suggests that strong regional diurnal variations exist for both AOD and aerosol
extinction profiles. Still, at present these conclusions are tentative, and will remain so until a
comprehensive analysis of the CATS calibration accuracy and stability is completed. These results
demonstrate the need for global aerosol measurements throughout the entire diurnal cycle to
improve visibility and particulate matter forecasts as well as studies focused on aerosol climate
applications.

**Author Contribution:**
Authors J. Zhang, J. S. Reid and L. Lee designed the study. L. Lee worked on data processing for
the project. J. E. Yorks guided L. Lee on data processing. The manuscript was written with inputs
from all coauthors.
**Acknowledgments:**
We acknowledge the support of ONR grant (N00014-16-1-2040) and NASA grant
(NNX17AG52G) for this study. L. Lee is also partially supported by the NASA NESSF
fellowship grant (NNX16A066H). J. S Reid's participation was supported by the Office of Naval
Research Code 322 and 33. We thank the NASA AERONET team for the AERONET data used
in this study. CATS and CALIOP data were obtained from the NASA Langley Research Center
Atmospheric Science Data Center. MODIS aerosol products were obtained from the NASA
Goddard Space Flight Center's MODIS Adaptive Processing System (MODAPS) site. We thank
Mark Vaughan and other two anonymous reviewers for their constructive suggestions and
comments.

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

Table 1. Descriptive statistical properties between collocated CATS and AERONET, CALIOP and Aqua MODIS AOD retrievals.
Here STDDEV indicates standard deviation of AOD and R-value represents the correlation coefficient.

| Sensor | No. of Points | Slope | R-value | Mean AOD | Median AOD | Max AOD | Min AOD | STDDEV | CATS Mean AOD | CATS Median AOD | CATS Max AOD | CATS Min AOD | CATS STDDEV |
|---|---|---|---|---|---|---|---|---|---|---|---|---|---|
| AERONET | 2240 | 0.56 | 0.65 | 0.088 | 0.054 | 0.98 | 0.001 | 0.103 | 0.099 | 0.058 | 1.31 | 0.0004 | 0.119 |
| MODIS Aqua | 3529 | 0.7 | 0.74 | 0.067 | 0.048 | 0.81 | 0.0004 | 0.07 | 0.07 | 0.053 | 1.76 | 0.002 | 0.075 |
| MODIS Terra | 2334 | 0.74 | 0.72 | 0.076 | 0.056 | 0.9 | 0.0013 | 0.081 | 0.084 | 0.065 | 1.13 | 0.0063 | 0.079 |
| CALIOP | 2762 | 0.74 | 0.74 | 0.089 | 0.063 | 1.01 | 0 | 0.102 | 0.092 | 0.065 | 1.1 | 0.0018 | 0.1 |


Table 2. Sensitivity study of descriptive statistical properties between collocated CATS and AERONET, CALIOP and Aqua MODIS
AOD retrievals by varying spatial and temporal collocation windows. Here STDDEV indicates standard deviation of AOD and R-
value represents the correlation coefficient.

| Collocation Thresholds *Spatial (30 min.)* | AERONET/CATS | | | AERONET | | | | | CATS | | | | |
|---|---|---|---|---|---|---|---|---|---|---|---|---|---|
| | No. of Points | Slope | R-value | Mean AOD | Median AOD | Max AOD | Min AOD | STDDEV | Mean AOD | Median AOD | Max AOD | Min AOD | STDDEV |
| 0.2° | 904 | 0.54 | 0.63 | 0.092 | 0.052 | 0.82 | 0.002 | 0.107 | 0.102 | 0.058 | 1.31 | 0.0004 | 0.124 |
| 0.4° | 2240 | 0.56 | 0.65 | 0.088 | 0.054 | 0.98 | 0.001 | 0.103 | 0.099 | 0.058 | 1.31 | 0.0004 | 0.119 |
| 0.8° | 5114 | 0.53 | 0.63 | 0.087 | 0.052 | 0.98 | 0.001 | 0.105 | 0.097 | 0.055 | 2 | 0.0004 | 0.125 |
| *Temporal (0.4° lat./lon.)* | No. of Points | Slope | R-value | Mean AOD | Median AOD | Max AOD | Min AOD | STDDEV | Mean AOD | Median AOD | Max AOD | Min AOD | STDDEV |
| 15 minutes | 1931 | 0.54 | 0.63 | 0.089 | 0.053 | 0.98 | 0.001 | 0.105 | 0.1 | 0.057 | 1.34 | 0.0004 | 0.123 |
| 30 minutes | 2240 | 0.56 | 0.65 | 0.088 | 0.054 | 0.98 | 0.001 | 0.103 | 0.099 | 0.058 | 1.31 | 0.0004 | 0.119 |
| 60 minutes | 2695 | 0.55 | 0.64 | 0.087 | 0.053 | 0.98 | 0.001 | 0.103 | 0.098 | 0.057 | 1.32 | 0.0004 | 0.118 |
| **Collocation Thresholds** *Spatial (30 min.)* | CALIOP/CATS | | | CALIOP | | | | | CATS | | | | |
| | No. of Points | Slope | R-value | Mean AOD | Median AOD | Max AOD | Min AOD | STDDEV | Mean AOD | Median AOD | Max AOD | Min AOD | STDDEV |
| 0.2° | 1948 | 0.73 | 0.76 | 0.088 | 0.063 | 1.15 | 0 | 0.104 | 0.092 | 0.065 | 1.12 | 0.0013 | 0.1 |
| 0.4° | 2762 | 0.74 | 0.74 | 0.089 | 0.063 | 1.01 | 0 | 0.102 | 0.092 | 0.065 | 1.1 | 0.0018 | 0.1 |
| 0.8° | 5070 | 0.80 | 0.74 | 0.089 | 0.063 | 0.94 | 0 | 0.099 | 0.093 | 0.066 | 1.61 | 0.0008 | 0.107 |
| *Temporal (0.4° lat./lon.)* | No. of Points | Slope | R-value | Mean AOD | Median AOD | Max AOD | Min AOD | STDDEV | Mean AOD | Median AOD | Max AOD | Min AOD | STDDEV |
| 15 minutes | 1392 | 0.76 | 0.77 | 0.09 | 0.063 | 0.95 | 0 | 0.104 | 0.092 | 0.066 | 1.1 | 0.0024 | 0.102 |
| 30 minutes | 2762 | 0.74 | 0.74 | 0.089 | 0.063 | 1.01 | 0 | 0.102 | 0.092 | 0.065 | 1.1 | 0.0018 | 0.1 |
| 60 minutes | 5602 | 0.74 | 0.75 | 0.09 | 0.063 | 1.4 | 0 | 0.104 | 0.093 | 0.066 | 1.55 | 0.0007 | 0.103 |
| **Collocation Thresholds** *Spatial (30 min.)* | MODIS Aqua/CATS | | | MODIS Aqua | | | | | CATS | | | | |
| | No. of Points | Slope | R-value | Mean AOD | Median AOD | Max AOD | Min AOD | STDDEV | Mean AOD | Median AOD | Max AOD | Min AOD | STDDEV |
| 0.2° | 2998 | 0.73 | 0.75 | 0.062 | 0.043 | 0.86 | 0.0004 | 0.073 | 0.07 | 0.052 | 1.74 | 0.003 | 0.075 |
| 0.4° | 3529 | 0.7 | 0.74 | 0.067 | 0.048 | 0.81 | 0.0004 | 0.07 | 0.07 | 0.053 | 1.76 | 0.002 | 0.075 |
| 0.8° | 4107 | 0.67 | 0.74 | 0.07 | 0.053 | 0.79 | 0.0004 | 0.066 | 0.071 | 0.053 | 1.71 | 0.003 | 0.073 |
| *Temporal (0.4° lat./lon.)* | No. of Points | Slope | R-value | Mean AOD | Median AOD | Max AOD | Min AOD | STDDEV | Mean AOD | Median AOD | Max AOD | Min AOD | STDDEV |
| 15 minutes | 1814 | 0.61 | 0.71 | 0.064 | 0.048 | 0.82 | 0.0004 | 0.067 | 0.069 | 0.052 | 1.76 | 0.003 | 0.078 |
| 30 minutes | 3529 | 0.70 | 0.74 | 0.067 | 0.048 | 0.81 | 0.0004 | 0.07 | 0.07 | 0.053 | 1.76 | 0.002 | 0.075 |
| 60 minutes | 6490 | 0.78 | 0.76 | 0.069 | 0.049 | 1.21 | 0.0004 | 0.076 | 0.072 | 0.054 | 1.76 | 0.003 | 0.074 |


Table 3. CALIOP and CATS mean AODs / AOD standard deviations for regions as highlighted in Figure 6 and globally between +/-
52° latitude.

| Region | Latitude | Longitude | Mean CATS AOD (DJFMAM/JJASON) | Mean CALIOP AOD (DJFMAM/JJASON) | Mean CATS Standard Deviation (DJFMAM/JJASON) | Mean CALIOP Standard Deviation (DJFMAM/JJASON) |
|---|---|---|---|---|---|---|
| Global | 52°S-52°N | 180°W-180°E | 0.09/0.10 | 0.09/0.09 | 0.037/0.039 | 0.036/0.034 |
| India | 7.5°N - 32.5°N | 65°E - 85°E | 0.22/0.26 | 0.22 /0.28 | 0.068/0.072 | 0.072/0.078 |
| Africa - North | 2.5°N - 22.5°N | 35°W - 20°E | 0.25/0.24 | 0.30 /0.25 | 0.062/0.064 | 0.075/0.067 |
| Africa - South | 17.5°S - 2.5°N | 0° - 30°E | 0.12/0.20 | 0.15 /0.13 | 0.037/0.048 | 0.038/0.038 |
| Middle East | 12.5°N - 27.5°N | 35°E - 50°E | 0.23/0.35 | 0. 26/0.35 | 0.076/0.099 | 0.082/0.091 |
| China | 27.5°N - 37.5°N | 110°E - 120°E | 0.20/0.17 | 0.21 /0.16 | 0.061/0.056 | 0.074/0.060 |


Table 4. Geographic ranges, height above ground level of maximum extinction, diurnal extinction range at height of maximum
extinction, and time (local) of peak extinction for the boxed red regions in Figure 6 and vertical profiles shown in Figures 12 and 13.

| DJFMAM/JJASON | | | | | |
|---|---|---|---|---|---|
| Region | Latitude | Longitude | Height AGL (m) of Max. Extinction | Extinction Range (km$^{-1}$) at Height AGL of Max. Extinction | Time of Peak Extinction at Height AGL of Max. Extinction |
| India | 7.5°N - 32.5°N | 65°E - 85°E | 180/360 | 0.099-0.136/0.135-0.163 | 0 am/6 am |
| Africa - North | 2.5°N - 22.5°N | 35°W - 20°E | 420/420 | 0.107-0.121/0.082-0.113 | 6 am/6 am |
| Africa - South | 17.5°S - 2.5°N | 0° - 30°E | /420 | /0.092-0.126 | /6 am |
| Middle East | 12.5°N - 27.5°N | 35°E - 50°E | 180/240 | 0.075-0.121/0.086-0.156 | 0 am/0 am |
| China | 27.5°N - 37.5°N | 110°E - 120°E | 180/240 | 0.098-0.148/0.086-0.132 | 6 am/6 am |


 **Figure Captions**


**Figure 1.** Collocated AERONET 1020 nm AOT vs. CATS 1064 nm AOD a) without CATS QA
applied, and b) with CATS QA applied.
**Figure 2.** Collocated MODIS C6.1 a) Terra and b) Aqua interpolated 1064 nm AOD vs. CATS
1064 nm AOD with CATS QA applied.
**Figure 3.** Collocated CALIOP 1064 nm AOD vs. CATS 1064 nm AOD with CATS QA applied
for a) both day and night, b) nighttime over-land, c) nighttime over-water, d) daytime over-land,
e) daytime over-water.
**Figure 4:** CATS 1064 nm AOD a) as a function of local time for the globe, and b) as a function
of local time for areas south of -25 degrees. The difference between CATS 1064 nm AOD and
AERONET 1020 nm AOD as a function of local time is shown in c). The mean is represented
by the blue line, while the median is the green line.
**Figure 5.** CATS and CALIOP vertical profiles of 1064 nm extinction for a) all profiles, b)
daytime only, c) nighttime only, d) over-water, and e) over land. f) shows the difference between
CATS and CALIOP mean 1064 nm extinction for all collocated profiles (5a) as a function of
height. Mean AOD values are as follows: for CATS: a) 0.094 , b) 0.091 , c) 0.098, d) 0.088, e)
0.119, and for CALIOP: a) 0.093, b) 0.092, c) 0.093, d) 0.084, e) 0.127.
**Figure 6.** Mean AOD (1064 nm) by season for a) DJFMAM CATS, b) JJASON CATS, c)
DJFMAM CALIOP, d) JJASON CALIOP, e) DJFMAM MODIS Aqua, and f) JJASON MODIS
Aqua. Red boxes indicate locations of regional vertical distributions in Figures 12 and 13.
**Figure 7.** Mean CATS AOD (1064 nm) by season for a) DJFMAM below 1 km AGL, b)
JJASON below 1 km AGL, c) DJFMAM 1-2 km AGL, d) JJASON 1-2 km AGL, e) DJFMAM
above 2 km AGL, and f) JJASON above 2 km AGL.
**Figure 8.** Seasonal Mean AOD (1064 nm) binned by every 6-hours for a) DJFMAM 0 UTC, b)
JJASON 0 UTC, c) DJFMAM 6 UTC, d) JJASON 6 UTC, e) DJFMAM 12 UTC, f) JJASON 12
UTC, g) DJFMAM 18 UTC, and h) JJASON 18 UTC.
**Figure 9.** Maximum minus minimum mean seasonal AOD (1064 nm) for a) DJFMAM, and b)
JJASON.
**Figure 10.** Global mean 6-hourly vertical profiles of CATS 1064 nm extinction for a) DJFMAM
all profiles, b) DJFMAM water profiles, c) DJFMAM not-water profiles, d) JJASON all profiles,
e) JJASON water profiles, f) JJASON not-water profiles. Mean AODs are as follows: a) 0.084,
b) 0.078, c) 0.098, d) 0.089, e) 0.082, and f) 0.102.
**Figure 11.** Global mean 6-hourly local time (0:00 am, 6:00 am, 12:00 pm and 6:00 pm) vertical
profiles of CATS 1064 nm extinction for a) DJFMAM all profiles, b) DJFMAM water profiles, c)
DJFMAM not-water profiles, d) JJASON all profiles, e) JJASON water profiles, f) JJASON not-
water profiles. Mean AODs are as follows: a) 0.080, b) 0.079, c) 0.095, d) 0.082, e) 0.081, and f)
826 0.105.


**Figure 12.** DJFMAM 6-hourly average (local time; 0:00 am, 6:00 am, 12:00 pm and 6:00 pm)
vertical profiles of CATS 1064 nm for locations shown in Figure 6a; a) Africa-North, b) Middle
East, c) India, and d) Northeast China.


**Figure 13.** JJASON  6-hourly average (local time; 0:00 am, 6:00 am, 12:00 pm and 6:00 pm)
vertical profiles of CATS 1064 nm for locations shown in Figure 6b; a) Africa-North, b) Africa-
South, c) Middle East, d) India, and e) Northeast China.


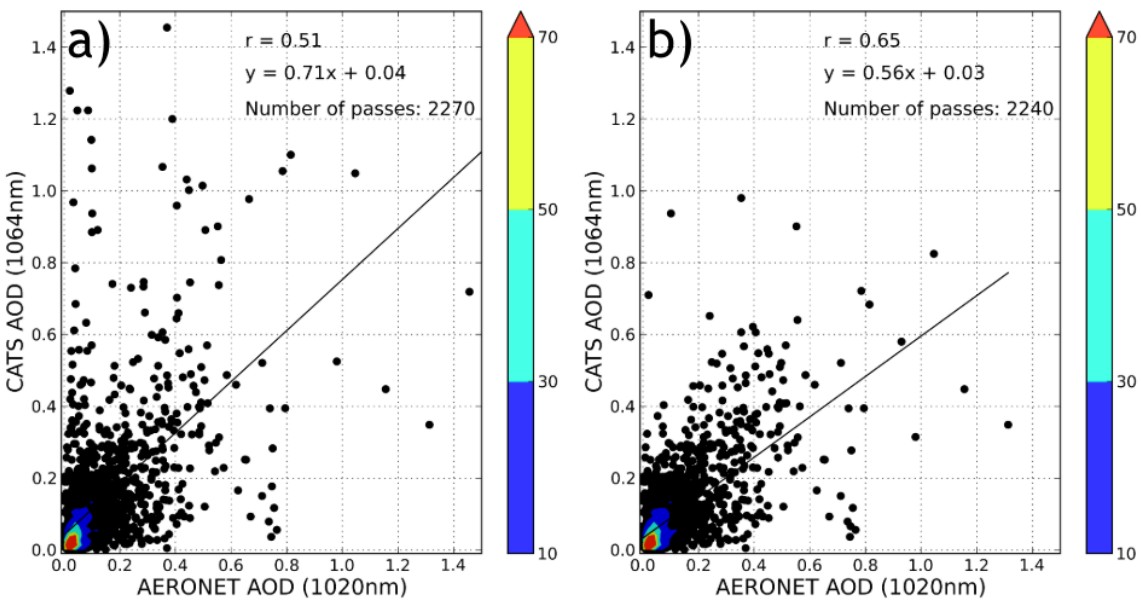


Figure 1. Collocated AERONET 1020 nm AOT vs. CATS 1064 nm AOD a) without CATS QA applied, and b) with CATS QA applied.

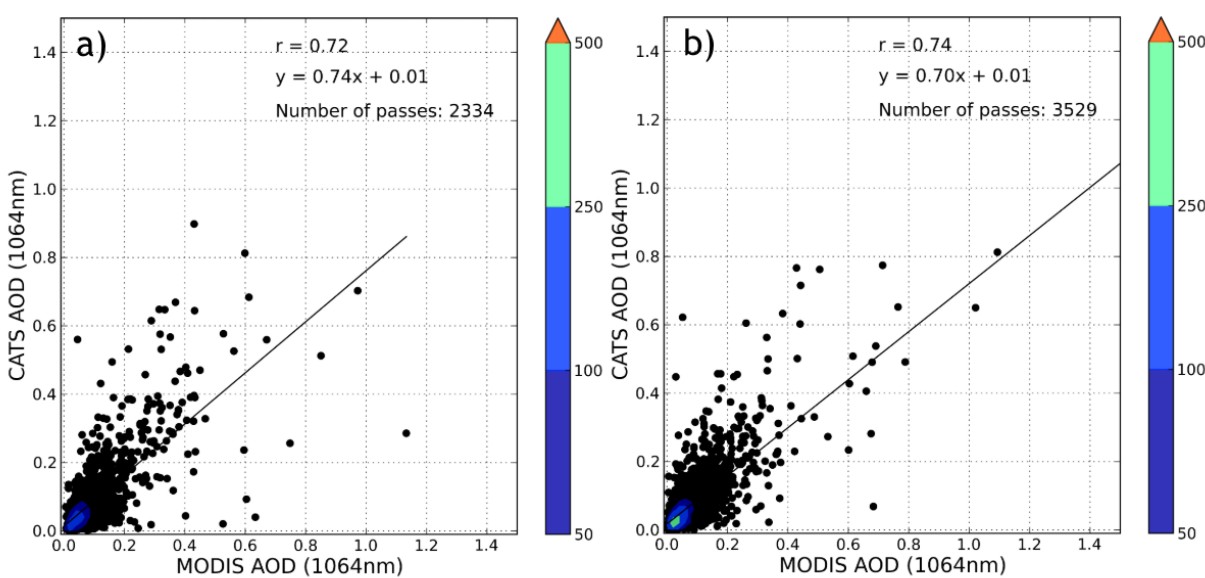


Figure 2. Collocated MODIS C6.1 a) Terra and b) Aqua interpolated 1064 nm AOD vs. CATS 1064 nm AOD with CATS QA applied.


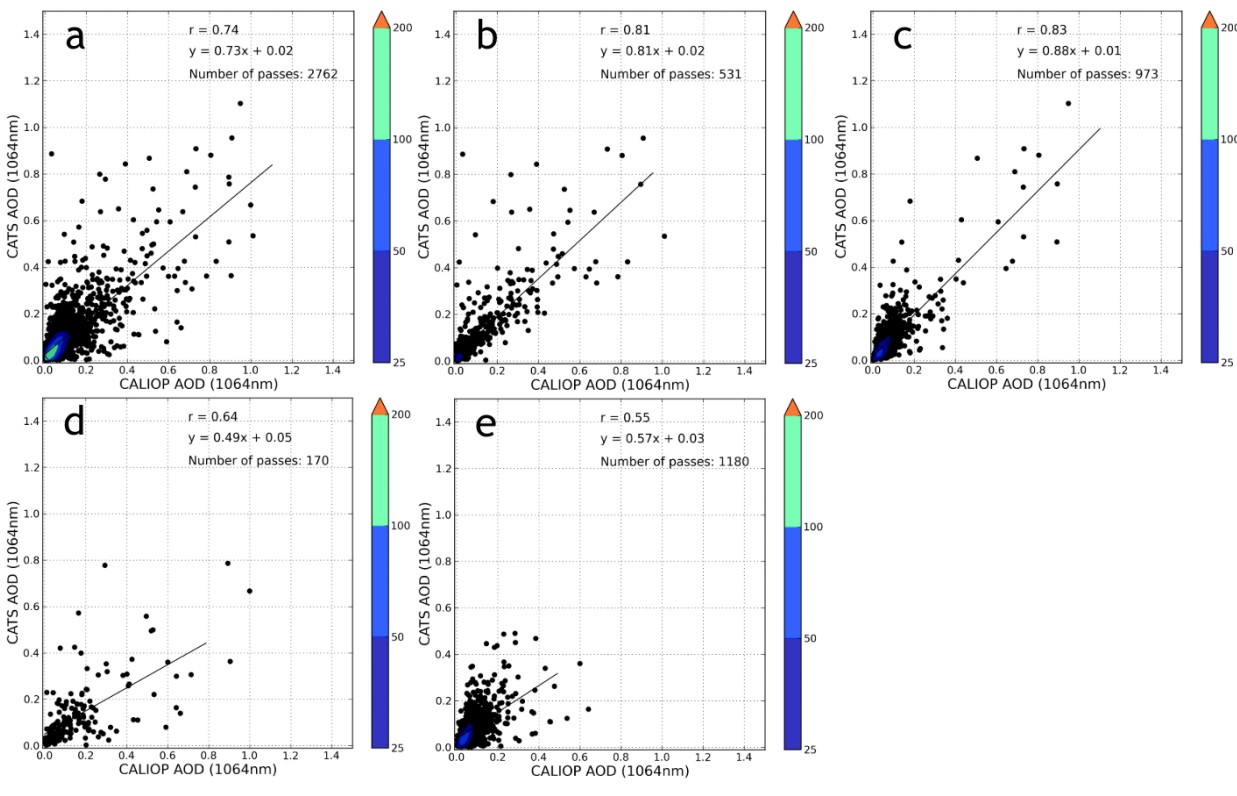

Figure 3. Collocated CALIOP 1064 nm AOD vs. CATS 1064 nm AOD with CATS QA applied for a) both day and night, b) nighttime over-land, c) nighttime over-water, d) daytime over-land, e) daytime over-water.

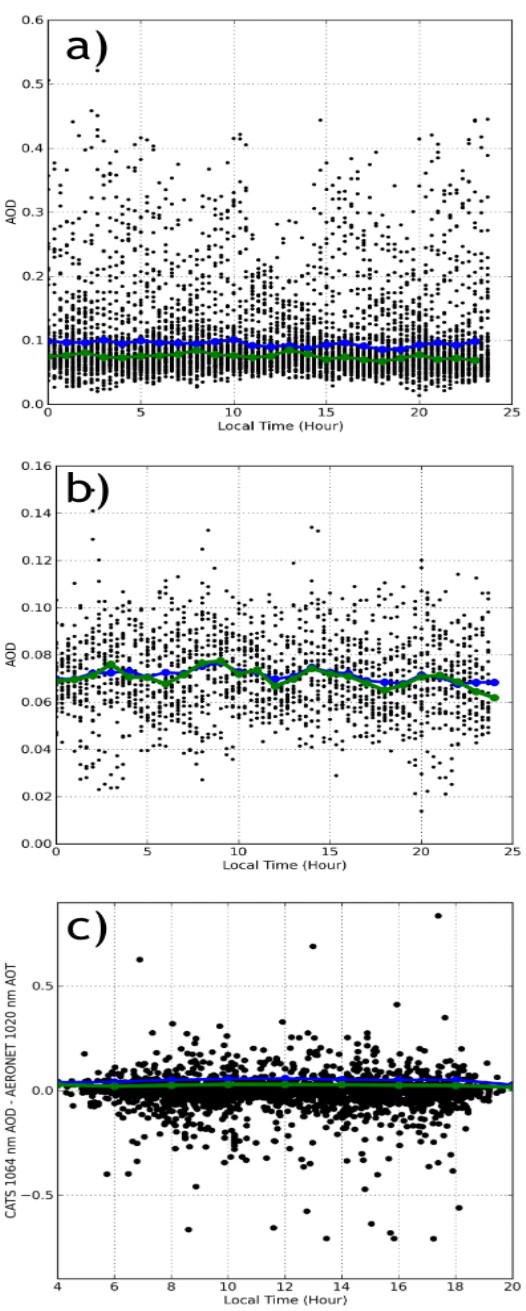


Figure 4. CATS 1064 nm AOD a) as a function of local time for the globe, and b) as a function of local time for areas south of -25 degrees. The difference between CATS 1064 nm AOD and AERONET 1020 nm AOD as a function of local time is shown in c). The mean is represented by the blue line, while the median is the green line.

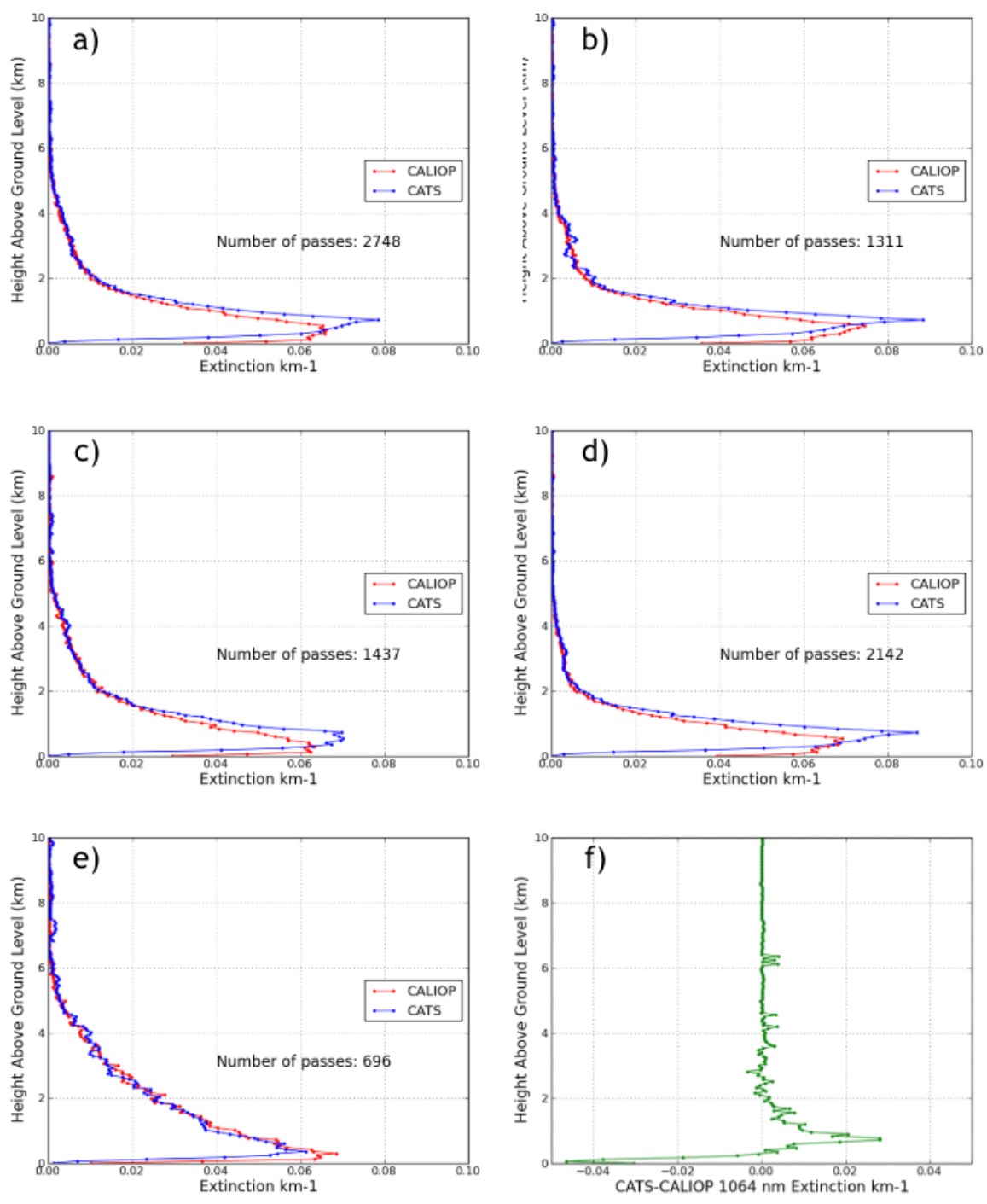


Figure 5. CATS and CALIOP vertical profiles of 1064 nm extinction for a) all profiles, b) daytime only, c) nighttime only, d) over-water, and e) over land. f) shows the difference between CATS and CALIOP mean 1064 nm extinction for all collocated profiles (5a) as a function of height. Mean AOD values are as follows: for CATS: a) 0.094 , b) 0.091 , c) 0.098, d) 0.088, e) 0.119, and for CALIOP: a) 0.093, b) 0.092, c) 0.093, d) 0.084, e) 0.127.

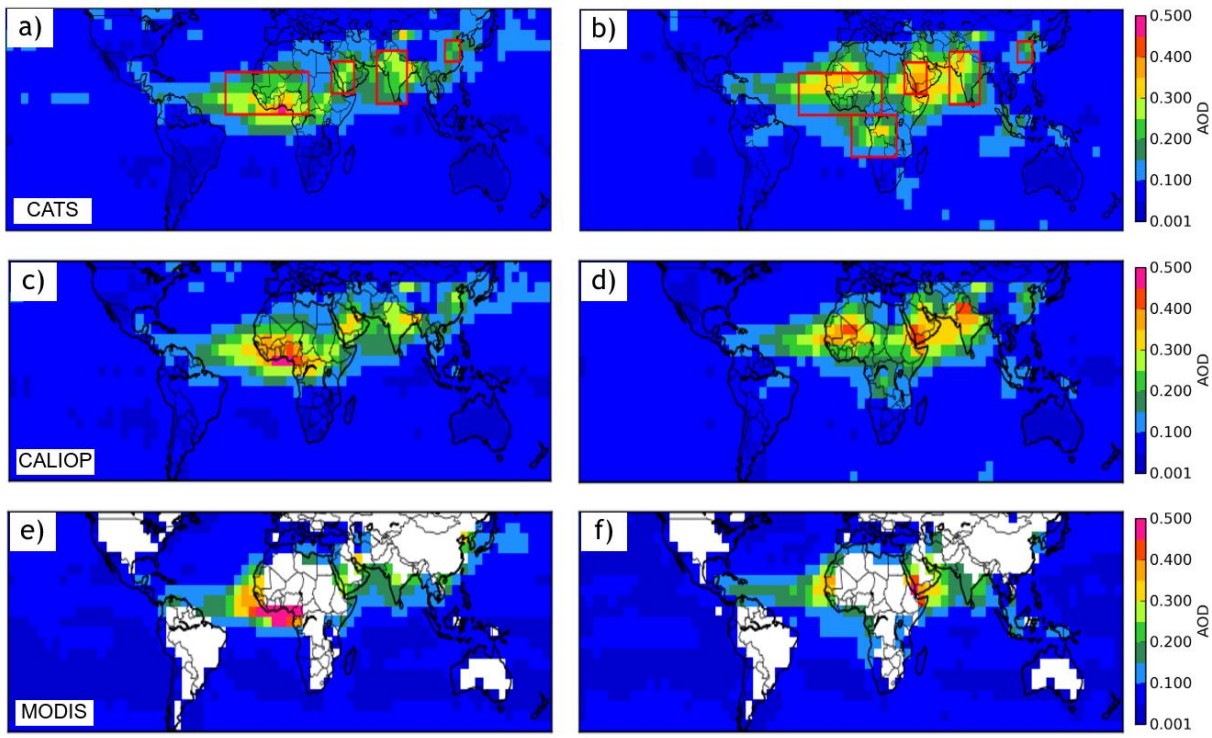



Figure 6. Mean AOD (1064 nm) by season for a) DJFMAM CATS, b) JJASON CATS, c) DJFMAM CALIOP, d) JJASON CALIOP, e) DJFMAM MODIS Aqua, and f) JJASON MODIS Aqua. Red boxes indicate locations of regional vertical distributions in Figures 12 and 13.






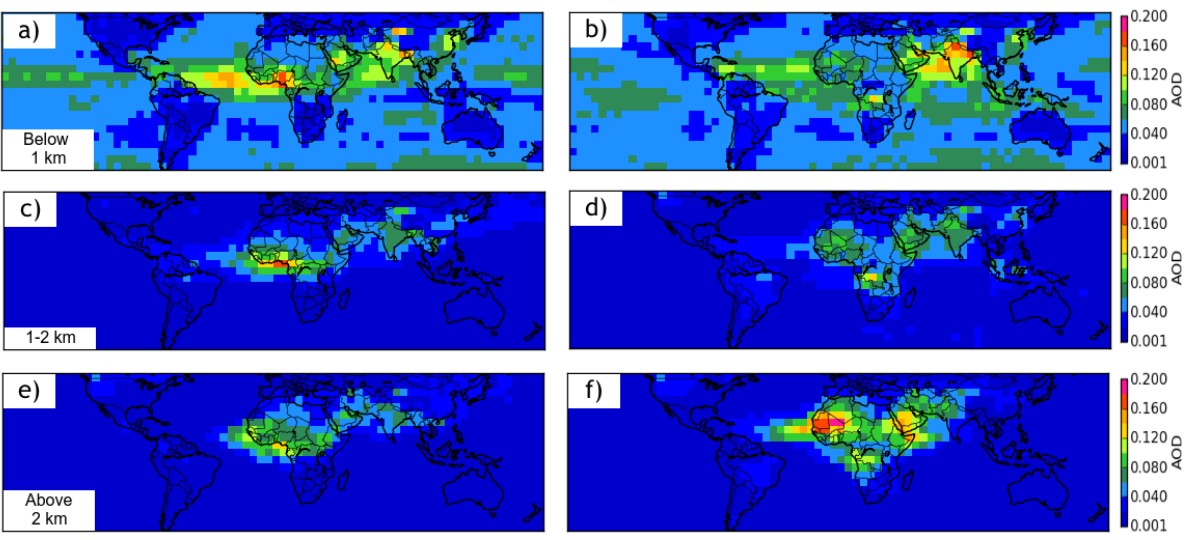


Figure 7. Mean CATS AOD (1064 nm) by season for a) DJFMAM below 1km AGL, b) JJASON below 1 km AGL, c) DJFMAM 1-2 km AGL, d) JJASON 1-2 km AGL, e) DJFMAM above 2 km AGL, and f) JJASON above 2 km AGL.

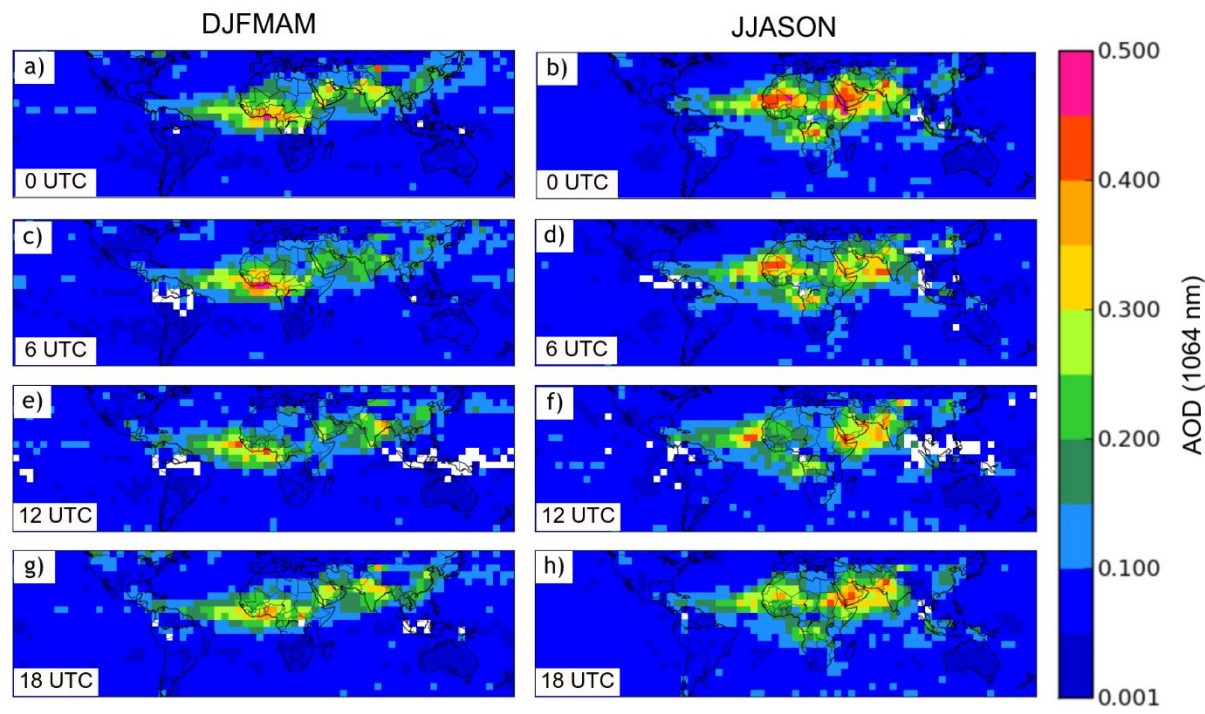


Figure 8. Seasonal Mean AOD (1064 nm) binned by every 6-hours for a) DJFMAM 0 UTC, b) JJASON 0 UTC, c) DJFMAM 6 UTC, d) JJASON 6 UTC, e) DJFMAM 12 UTC, f) JJASON 12 UTC, g) DJFMAM 18 UTC, and h) JJASON 18 UTC.

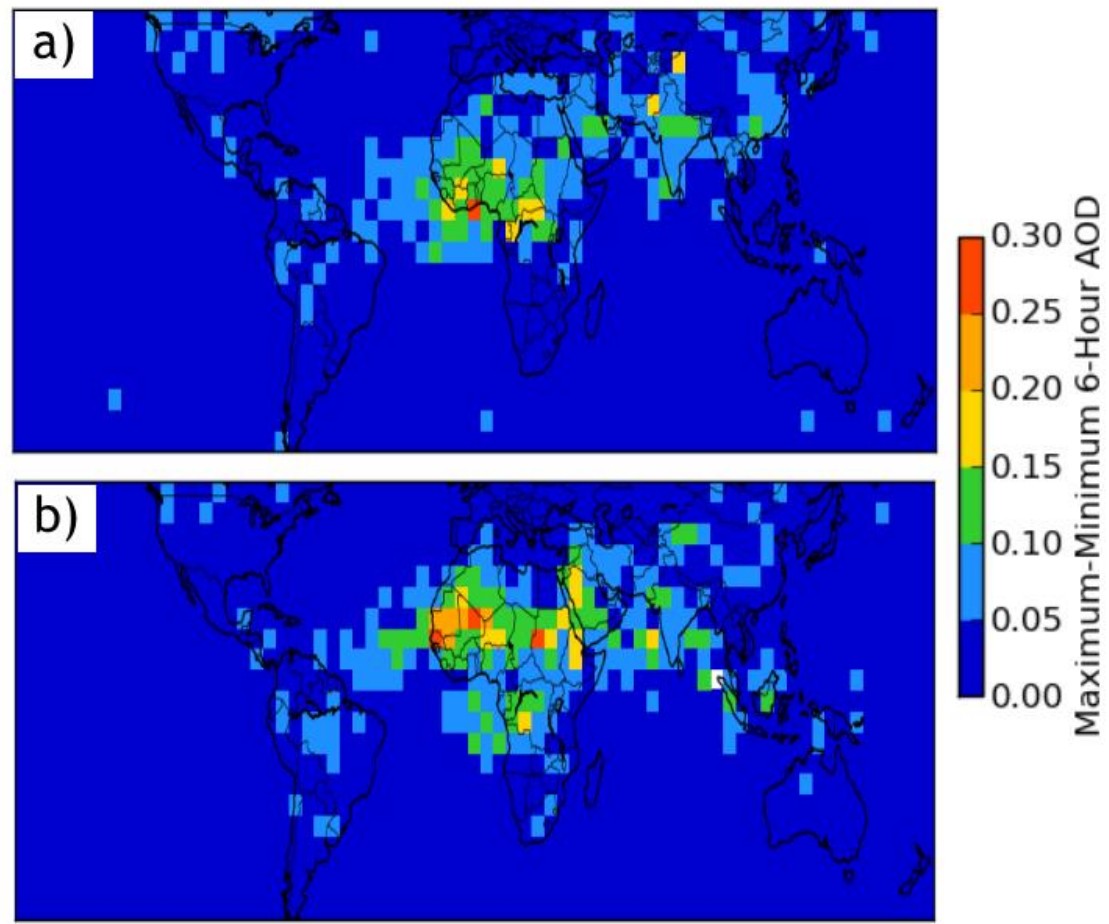

Figure 9.  Maximum minus minimum mean seasonal AOD (1064 nm) for a) DJFMAM, and b) JJASON.

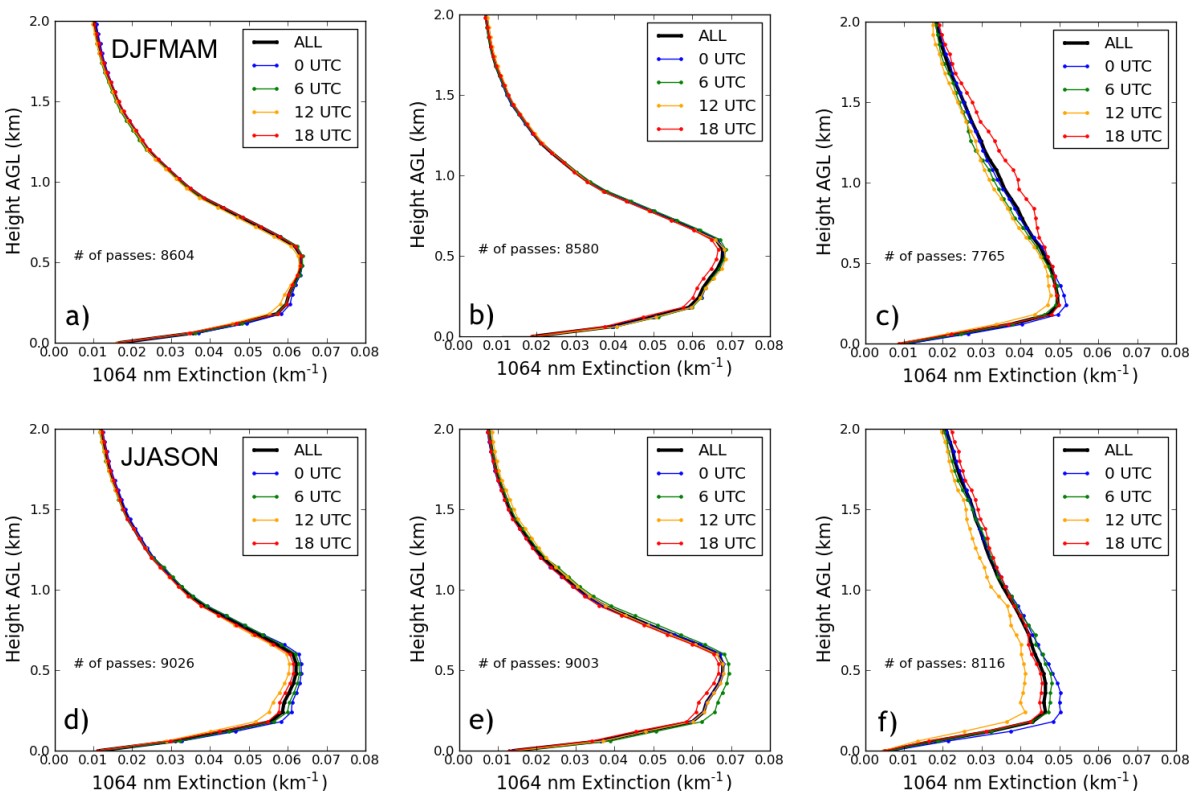


Figure 10. Global mean 6-hourly vertical profiles of CATS 1064 nm extinction for a) DJFMAM all profiles, b) DJFMAM water profiles, c) DJFMAM not-water profiles, d) JJASON all profiles, e) JJASON water profiles, f) JJASON not-water profiles. Mean AODs are as follows: a) 0.084, b) 0.078, c) 0.098, d) 0.089, e) 0.082, and f) 0.102.

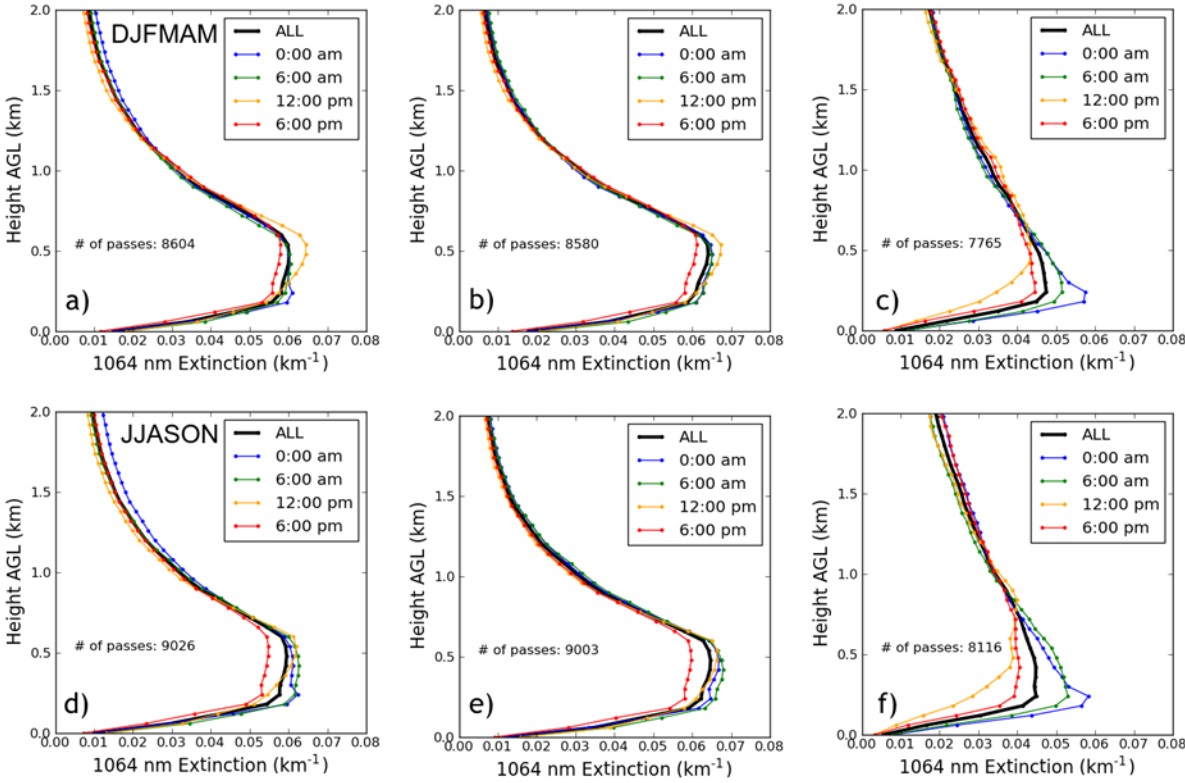


Figure 11. Global mean 6-hourly local time (0:00 am, 6:00 am, 12:00 pm and 6:00 pm) vertical profiles of CATS 1064 nm extinction for a) DJFMAM all profiles, b) DJFMAM water profiles, c) DJFMAM not-water profiles, d) JJASON all profiles, e) JJASON water profiles, f) JJASON not-water profiles. Mean AODs are as follows: a) 0.080, b) 0.079, c) 0.095, d) 0.082, e) 0.081, and f) 0.105.










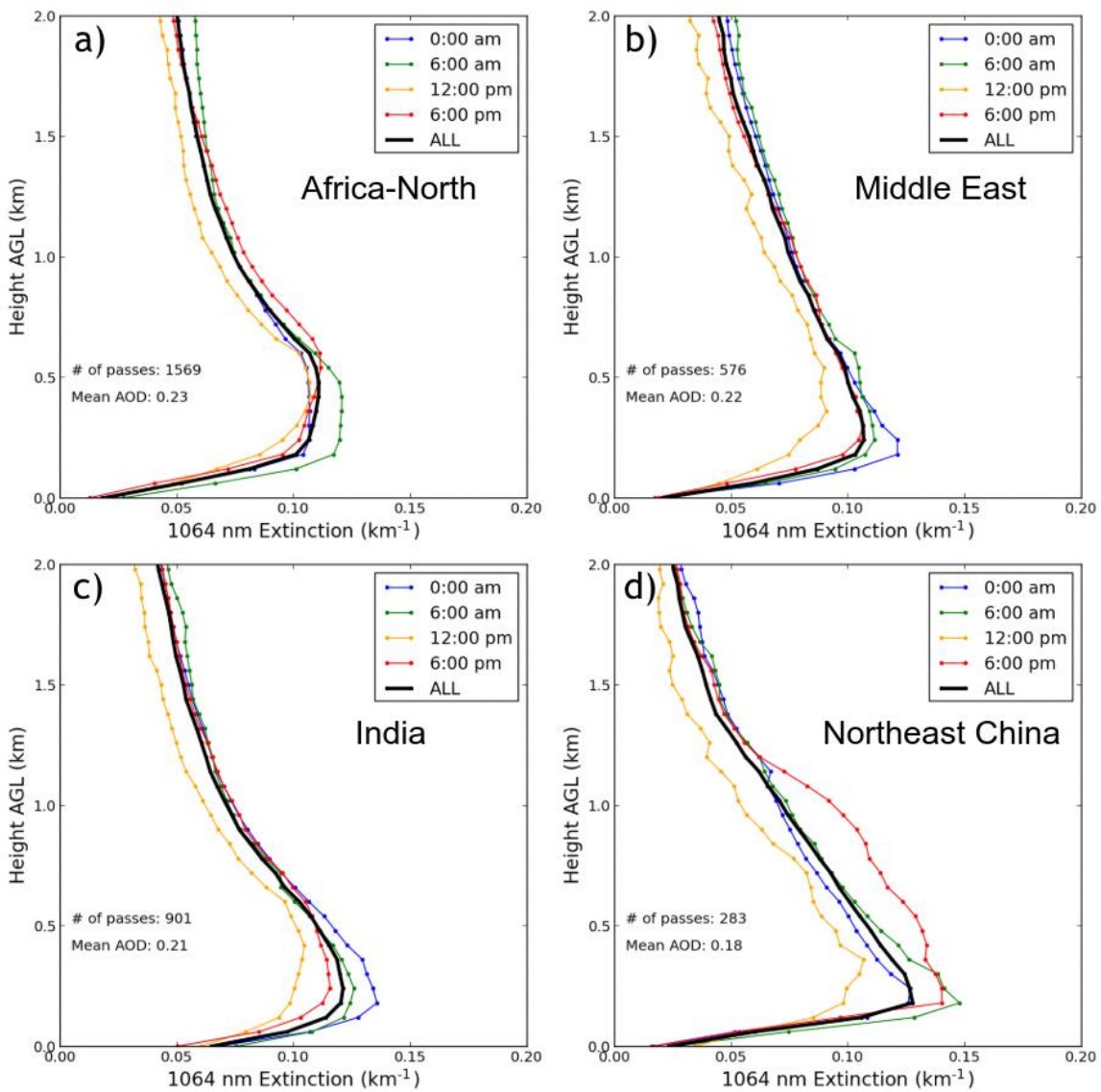


Figure 12. DJFMAM 6-hourly average (local time; 0:00 am, 6:00 am, 12:00 pm and 6:00
pm) vertical profiles of CATS 1064 nm for locations shown in Figure 6a; a) Africa-North,
b) Middle East, c) India, and d) Northeast China.






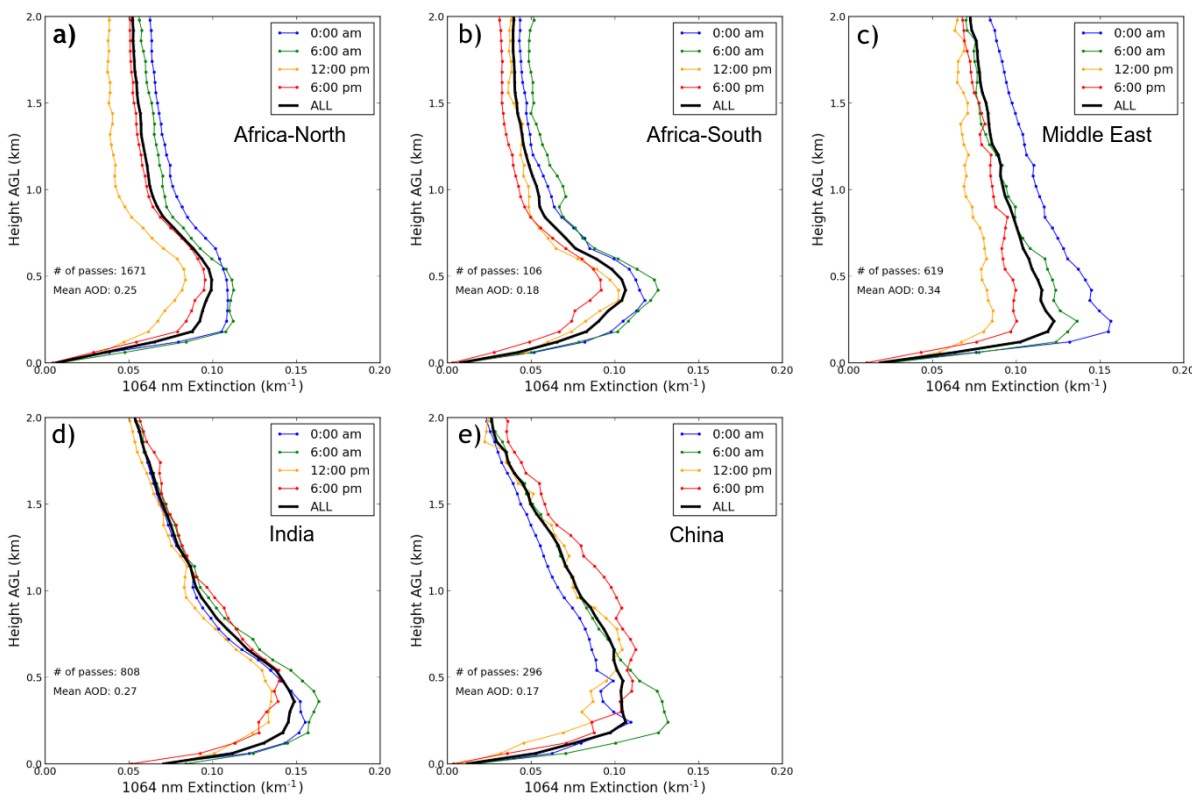


Figure 13.  JJASON  6-hourly average (local time; 0:00 am, 6:00 am, 12:00 pm and 6:00 pm) vertical profiles of CATS 1064 nm for locations shown in Figure 6b; a) Africa-North, b) Africa-South, c) Middle East, d) India, and e) Northeast China.

Appendix A:

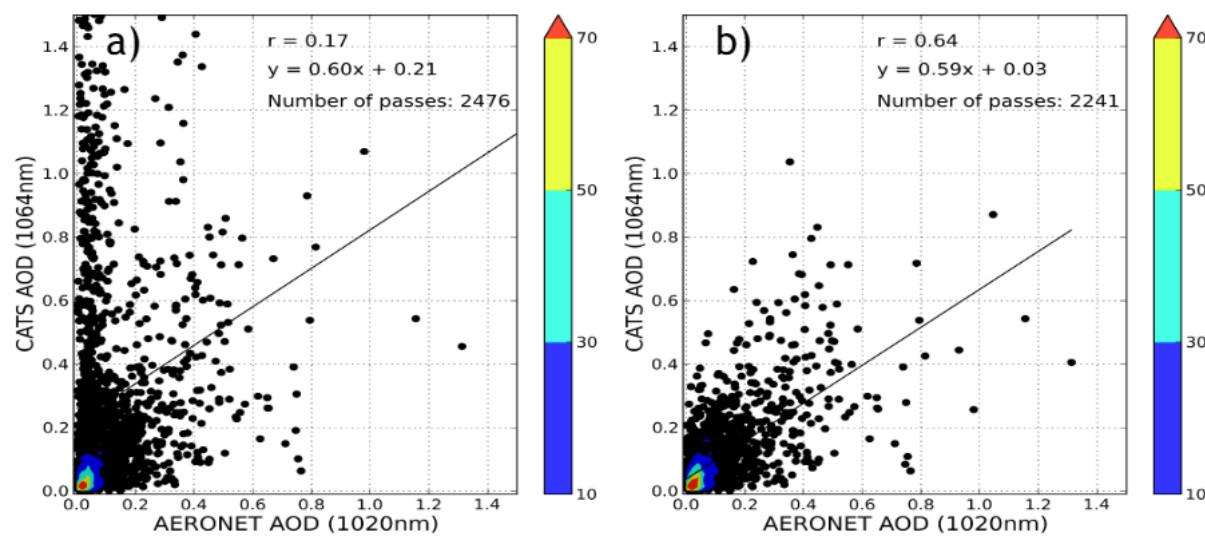

Figure A1. Collocated AERONET 1020 nm AOT vs. CATS 1064 nm AOD a) without CATS
QA applied, and b) with CATS QA applied.  CATS V2-01 aerosol products were used in
constructing this plot.