# Peer review of "1.0 Introduction"

_Atmospheric Chemistry and Physics, 2018_

## Referee Comment (RC1) · Anonymous Referee #1 · 26 Jan 2019

The manuscript named "Investigation of CATS aerosol products and application toward global diurnal variation of aerosols" by Lee et al. presents an intercomparison of the measurements of aerosol optical depth and mean profiles between CATS and other remote sensing sensors (AERONET, MODIS, and CALIOP) for a period of Feb. 2015 - Oct. 2017. This paper also discusses the aerosol diurnal variation patterns changing with different seasons and geographic regions. This manuscript presents an original data analysis of some significant instruments. The discussion and conclusions are sound and clear. Therefore, I recommend for publish after addressing some minor concerns.

Specific comments: Section 2, can you briefly describe the AOD measurement uncertainty of these instrument? P6, L134, it may be better to replace "increasing" with

"degrading". P8, L163, can you describe what constant value of that Angstrom exponent is used here without letting readers to look for that in Shi et al. paper? P12, L266-268, "A clear diurnal variation is found, with the peak mean AOD of 0.08 found around local noon and smaller AOD values of 0.06 found for both sunrise and sunset times." In Figure 4, look to me the AOD peak is located around 9AM local time, "before" the noon. Also, is this diurnal variation consistent with your expectation? Can you provide an explanation on why the AOD measured by CATS less than all other instruments suggested by Figure 1, 2, and 3?

---

## Referee Comment (RC2) · Anonymous Referee #2 · 28 Jan 2019

The authors use more than two years of CATS data to examine the diurnal cycles of the aerosol loading on global scale. Their results show that a strong peak at 6 am local time in aerosol extinction profile over North Africa during the June-November season. This finding is exciting and brand new. I would recommend this manuscript be published in ACP after a few minor changes.

(1) In Figure 5, there are some spikes above 2 km in the aerosol extinction vertical profiles seen in the CATS data, but not present in the CALOP data. Are they due to the cloud screening differences between CATS and CALIOP? (2) Line# 353-354, unlike CALIOP, MODIS Aqua aerosol products are only available in the early afternoon, but not in the early morning, since the algorithm only performs retrieval over daytime. (3) Line# 355-356, please add a sentence or two to briefly elaborate what aerosol above

cloud issues are as reported by Rajapashe et al., (2017). (4) Line# 358, please spell out "AGL". (5) The aerosol extinction at 1064 nm may not be as sensitive to the fine mode aerosols (such as smoke and urban pollutant aerosols) compared to the coarse mode aerosols (such as dust). The authors probably should add a few sentences to address this.
* * *

---

## Short Comment (SC1) · 13 Feb 2019

**Some comments on "Investigation of CATS aerosol products and application toward global diurnal variation of aerosols" by Logan Lee, Jianglong Zhang, Jeffrey S. Reid, and John E. Yorks**

Mark Vaughan (mark.a.vaughan@nasa.gov) and Stuart Young (stuart.young01@gmail.com)

This paper compares the aerosol extinction profiles and aerosol optical depths (AODs) retrieved by the CATS lidar with similar quantities retrieved by AERONET, MODIS, and CALIOP. To our knowledge, this is the first ever in-depth assessment of satellite-derived AODs measured/retrieved in the near-infrared, and thus should be of great interest to several different groups in the aerosol research community. Overall, the authors have done a good job in bringing multiple analyses together. However, we find several places where additional analyses are warranted and where more in-depth discussions will help strengthen the final manuscript.

**General Remarks**

When filtering the extinction coefficients retrieved from the CATS and CALIOP measurements, the authors say that candidate extinction coefficients were constrained to a "nominal range" of 0 to 1.25 $km^{-1}$, and that "all near zero negative values" are set to zero (page 6, lines 114–119). Presumably these "near zero negative values" that were set to zero were not actually removed from consideration, but instead were included in subsequent data averaging operations (the writing in this section is not sufficiently clear on this point). Changing negative values to zeros prior to averaging is not statistically valid, as it guarantees high biases in the estimated mean values. Reporting these high biases erroneously improves the comparisons of lidar-derived optical depths with those obtained by other sensors. To avoid this, all CATS and CALIOP mean values should be correctly recalculated before the final version of this manuscript is published.

While the main body of the text emphasizes the correlations between the CATS retrievals and the other data sets (e.g., lines 186–208), the authors do not provide any quantitative statements about the magnitudes of the CATS AODs or the differences between the different AOD estimates. Given that this paper is (to our knowledge) the first ever in-depth look at 1064 nm AOD, tables showing global and regional mean values and quantifying the CATS AOD estimates relative to the other sensors would add significantly to the value delivered by this paper. Profiles of the relative CATS-CALIOP extinction coefficient differences (i.e., (CATS(z) - CALIOP(z)) / CALIOP(z)) would be especially interesting.

In section 3.1.1., CATS observations are compared with other observations made within ±30 mins and ±0.4 degrees. For aerosols, this is probably not too much of a problem a lot of the time, but we have seen numerous cases where there can be large differences in the scenes being observed (e.g., see Omar et al., 2013: "In 45% of the coincident instances CALIOP and AERONET do not agree on the cloudiness of the scenes."). For AERONET, the comparisons may be improved by imposing another criterion, i.e., that the AERONET AODs made at the closest times preceding and following the CATS observations not vary by more than x%. A similar filter for potential spatial differences could include wind speed and direction (e.g., Lopes et al., 2013) and the spatial separations of the AERONET sites and the CATS observations. (This is likely to be quite a bit messier.)

While the authors point out a number of differences between the CATS retrievals and those derived from other sensors, they typically do not attempt to identify the causes of these differences. For example, based on the scaling factors in the linear regressions, the CATS AODs are lower than all of the AODs with which they are being compared (i.e., AERONET in Figure 1, MODIS in Figure 2, and CALIOP in Figure 3). This is perhaps not surprising for the AERONET and MODIS comparisons, but the cause for the CATS-CALIOP differences is not as obvious. Differences between CALIOP and MODIS at visible wavelengths are frequently explained by CALIOP's low daytime detection sensitivity and the missed detection of some of the vertical extent of the aerosol layer (e.g., Kim et al., 2017 and Toth et al., 2018). Can the authors enumerate the possible causes that would explain the disparities between CATS and CALIOP?

Furthermore, given the lower CATS AODs shown in Figure 2, it's surprising to see that the CATS extinctions coefficients shown in Figure 5 are typically larger than CALIOP at all altitudes, and that the closest agreement is over land (where CATS slightly underestimates CALIOP at lower altitudes). Again, some discussion of the possible causes of this paradox would be welcome.

The results shown in Figure 5 are a prime candidate for further investigation into the underlying causes of the differences. Except for the over land case, CATS extinction profiles consistently and significantly overestimate CALIOP extinction profiles. It seems that there are four likely suspects in causing this (always keeping in mind that that all four could be collaborating in various nefarious ways to bring this about): layer detection, cloud-aerosol discrimination (including inadequate boundary layer cloud clearing), lidar ratio selection, and calibration. Of these four, the easiest to investigate (at least at a superficial level) is lidar ratio. The table below shows the default lidar ratios assigned by each instrument.

| Aerosol Type | CATS | CALIOP |
|---|---|---|
| Dust | 40 sr | 44 sr |
| Dust mixture [a] | 40 sr | N/A |
| Polluted dust [a] | N/A | 48 sr |
| Dusty marine [a] | N/A | 37 sr |
| Marine | 25 sr | 23 sr |
| Clean/background | 35 sr | 30 sr |
| Polluted continental | 35 sr | 30 sr |
| Smoke | 40 sr | 30 sr |
| Volcanic [b] | 35 sr | 44 sr |

(a)  CATS identified dust mixtures over land and water; CALIOP identifies 'polluted dust' over land only and 'dusty marine' over water only.

(b)  For CATS, all aerosol above 10 km is classified as volcanic. For CALIOP, volcanic aerosol is identified in the stratosphere only.

Since the CATS marine lidar ratio is large than the CALIOP marine lidar ratio, and the CATS dust mixture lidar ratio is larger than the CALIOP dusty marine lidar ratio (and CATS smoke and polluted continental lidar ratios are greater than their CALIOP counterparts as well), then, *all other things being equal*, one should expect the CATS over-ocean extinction profiles to be uniformly larger than the CALIOP extinction profiles. (But are all other things actually equal?)

The case is less clear over land. But since the CATS dust lidar ratio is less than the CALIOP dust lidar ratio and the CATS dust mixture lidar ratio is less than the CALIOP polluted dust lidar ratio, if we assume that the over-land aerosols detected in this study are dominated by dust (which might not be a bad assumption?), then perhaps the over-land profile comparison makes sense too. (All other things being equal, that is…)

The CATS extinction profiles shown in Figures 5 and 10 peak at altitudes some hundreds of meters higher than do CALIOP's, except over land. While CALIOP's profiles show almost no roll off until about the last range bin above the surface, the CATS profiles start dropping off below about 500 m, or at approximately 8 to 9 range bins above the surface. What is happening here? Is CATS altitude registration and/or surface detection the culprit? Or is the cloud filter too aggressive in the boundary layer (i.e., are strongly scattering aerosols being misclassified as clouds)? Irrespective of the underlying cause(s), is this behavior a major source of AOD differences between CATS and CALIOP?

The seasonal maps (Figure 6) show that the CALIOP AODs exceed those of CATS over the Arabian Peninsula, and to a smaller degree over the African region bordering the Gulf of Guinea. Can this also be explained by differences in lidar ratio selection, or are there other factors at work?

**Specific Comments**

page 4, line 85: provide a reference for "Feature Type Score"

page 5, line 107: did the authors also consider potential sources of bias errors; e.g., unusually large or small calibration coefficients, or large values of overlying integrated attenuated backscatter?

page 5, line 113: "Extinction_Coefficient_Uncertainty_1064_Fore_FOV $\leq 10$ km$^{-1}$"; despite the heritage from Campbell et al. (2012), using relative uncertainties still makes much, much more sense. Given the noise in the CATS daytime measurements, an uncertainty threshold of 10 km$^{-1}$ might be reasonable for an estimated extinction coefficient of 1 km$^{-1}$. However, for the substantially smaller extinction coefficients (e.g., 0.01 km$^{-1}$ to 0.1 km$^{-1}$) that make up a very large majority of the measurements, an uncertainty threshold of 10 km$^{-1}$ seems prohibitively large.

page 6, line 128: distinguish between laser spot size (~70 m) and receiver footprint diameter at the Earth's surface (~90 m).

page 6, line 129: say which version of the CALIPSO data products was used (version 4.1, right?)

page 7, line 137: "signal-to-noise", not "single to noise"

page 7, line 148: "Atmospheric_Volume_Description = 3 (aerosol only)"; note that in the CALIPSO version 4.1 data products, 3 indicates tropospheric aerosols and 4 indicates stratospheric aerosols. Were stratospheric aerosols excluded accidentally or deliberately? (Previous versions of the CALIPSO data products did not differentiate between tropospheric and stratospheric aerosols. In these earlier products, requiring the atmospheric volume description to equal 3 would correctly identify all aerosol data.) If accidentally, please correct the calculations. If deliberately, please explain why.

page 8, line 163:    logarithmic interpolation, correct?  Also, please state the actual value of the Ångström exponent given by Shi et al.

page 8, line 170:    while "AERONET data are considered as the ground truth for evaluating CATS retrievals", it should be noted that there are very few AERONET sites in remote oceans.  Do MODIS retrievals substitute as the gold standard in these places?

page 9, line 186–187:    some discussion on the rationale for the choices of ±0.4° and ±30 minutes would be helpful in evaluating the strength of the comparisons.

page 9, line 193:    how frequently do "profiles with all retrieval fill values" occur in the CATS data set?

page 9, line 194:    as a rule of thumb, how close to sunrise and sunset can reliable AERONET measurements be obtained?

page 11, line 244:    The authors say, "using over land (ocean) daytime data only, for a total of 171 (1207) collocated pairs."   Here we echo the remarks of an anonymous reviewer commenting on a paper for which one of us (Mark Vaughan) is a coauthor (see https://doi.org/10.5194/acp-2018-1090-RC1).

*Way back in 2010 Prof. Robock pleaded with us to end this misuse of parentheses [Robock, A. (2010), Parentheses are (are not) for references and clarification (saving space), Eos Trans. AGU, 91(45), 419–419, doi:10.1029/2010EO450004].  My understanding is that one of the publishers in our field has specifically written it out of their style guide. I read pretty widely and the only genre of writing where I have experienced this application of parentheses is in the atmospheric sciences journals. I hope the authors will consider rewriting this sentence.*

page 11, line 245:    The authors say, "daytime data from both CALIOP and CATS are expected to be nosier due to solar contamination".  While this is true, the day-night differences at 1064 nm are very different for the two lidars.  CATS daytime SNR is substantially worse than CATS nighttime SNR, whereas CALIOP daytime SNR is only marginally worse than CALIOP nighttime SNR.  The primary reason for this is that CALIOP 1064 nm detector is an avalanche photodiode for which the dark counts contribute substantial amounts of noise irrespective of the external lighting conditions.  Moreover, while CATS 1064 nm nighttime SNR is much higher than CALIOP 1064 nm nighttime SNR, for daytime measurements the CALIOP SNR is higher.  This should be explained in greater detail in a forthcoming CATS calibration paper.

page 12, line 260:    "it is speculated".  Who's doing this speculating?  If it's the authors, then come right out and say so!

page 14, line 311:    The authors say, "the shapes of the CATS and the CALIOP nm extinction vertical profile are very similar for all three cases".  This qualitative assessment would be much more meaningful if it was augmented by a set of quantitative metrics (e.g., profiles of (CATS(z) - CALIOP(z)) / CALIOP(z), with error bars to indicate the magnitude of the variability in the ratios).

page 18, line 405:    The authors say, "nighttime retrievals from CATS *are considered to be less noisy* than daytime" (emphasis added).  This sentence suggests that there might be some debate about day versus night noise magnitudes.  There is no such debate.  The fact is that

"nighttime retrievals from CATS *are significantly and demonstrably less noisy* than daytime retrievals".

page 23, lines 514–517:    The authors' conclusions reinforce the conventional wisdom. However, we think it's important to emphasize that at present these conclusions are highly tentative, and will remain so until a comprehensive analysis of the CATS calibration accuracy and stability is completed.

**References**

Kim, M.-H. et al., 2017: Quantifying the low bias of CALIPSO's column aerosol optical depth due to undetected aerosol layers, *J. Geophys. Res. Atmos.*, **122**, 1098–1113, doi:10.1002/2016JD025797.

Lopes, F. et al., 2013: Evaluating CALIPSO's 532nm lidar ratio selection algorithm using AERONET sun photometers in Brazil, *Atmos. Meas. Tech.*, **6**, 3281–3299, doi:10.5194/amt-6-3281-2013.

Omar, A. et al., 2013: CALIOP and AERONET aerosol optical depth comparisons: One size fits none, *J. Geophys. Res. Atmos.*, **118**, 4748–4766, doi:10.1002/jgrd.50330.

Toth, T. D. et al., 2018: Minimum Aerosol Layer Detection Sensitivities and their Subsequent Impacts on Aerosol Optical Thickness Retrievals in CALIPSO Level 2 Data Products, *Atmos. Meas. Tech.*, **11**, 499–514, doi:10.5194/amt-11-499-2018.

---

## Author Response (AR1)

**Reviewer 1:**

Comments: The manuscript named "Investigation of CATS aerosol products and application toward global diurnal variation of aerosols" by Lee et al. presents an inter-comparison of the measurements of aerosol optical depth and mean profiles between CATS and other remote sensing sensors (AERONET, MODIS, and CALIOP) for a period of Feb. 2015 -Oct. 2017. This paper also discusses the aerosol diurnal variation patterns changing with different seasons and geographic regions. This manuscript presents an original data analysis of some significant instruments. The discussion and conclusions are sound and clear. Therefore, I recommend for publish after addressing some minor concerns.

Response: We thank the reviewer for his/her suggestions, comments and encouragement.

Comments: Specific comments: Section 2, can you briefly describe the AOD measurement uncertainty of these instrument?

Response: This is a great question. Most validation and uncertainties analysis efforts of satellite AOD retrievals are focus on visible channels. To our knowledge, uncertainties in AOD retrieval at 1064 nm, both from passive and active sensors, are less studied. Just as suggested from the comments from Mark Vaughan and Stuart Young (Short comment for this paper), this paper might be among the first to go deep into AOD retrievals at 1064 nm, channel. We were not able to find papers to address uncertainties in AOD retrievals at 1064 nm, although there are papers that do show comparisons between CALIOP and AERONET AOD at 1064 nm (Omar et al., 2013).

*Omar, A. H., D. M. Winker, J. L. Tackett, D. M. Giles, J. Kar, Z. Liu, M. A. Vaughan, K. A. Powell, and C. R. Trepte (2013), CALIOP and AERONET aerosol optical depth comparisons: One size fits none, J. Geophys. Res. Atmos., 118, 4748–4766, doi:10.1002/jgrd.50330.*

We have added the following discussion in the text: "Note that most evaluation efforts for passive- and active-based AOD retrievals are focused on the visible spectrum and the performance of AOD retrievals at the 1064 nm channel is less explored. "

Comments: P6, L134, it may be better to replace "increasing" with "degrading".

Response: Done

Comments: P8, L163, can you describe what constant value of that Angstrom exponent is used here without letting readers to look for that in Shi et al. paper?

*Response:* We apologize for the confusion. The Angstrom exponent values are computed using instantaneous retrievals. We have revised the text to avoid confusion.

"Here we assume the angstrom exponent value, computed using instantaneous AOD retrievals at the 860 and 1240 nm, remains the same for the 860 to 1064 nm wavelength range, similar to what has been suggested by Shi et al., (2011; 2013). "

Comments: P12,L266-268, "A clear diurnal variation is found, with the peak mean AOD of 0.08 found around local noon and smaller AOD values of 0.06 found for both sunrise and sunset times." In Figure 4, look to me the AOD peak is located around 9AM local time, "before" the noon. Also, is this diurnal variation consistent with your expectation?

*Response:* Thanks for the suggestion. We have revise the sentence to "with the mean AOD values of 0.07-0.08 found between late morning and early afternoon and smaller AOD values of 0.06 found for both sunrise and sunset times"

Comments: Can you pro-vide an explanation on why the AOD measured by CATS less than all other instruments suggested by Figure 1, 2, and 3?

Response: We assume that the reviewer is referring to the slope of the regressions in Figures 1-3. Slopes in linear regressions can often be biased by outliers. In Figure 6, which are spatial plots of AODs from CALIOP and CATS, differences are less noticeable for the DJFMAM season. For the JJASON season, CATS AODs are lower at certain regions (Middle East, India, and North Africa) and higher over other regions (South Africa). The cause of those discrepancies, however, is unclear to us. To really explore the issue, it deserves a paper of its own. Thus, we leave this topic to a future paper.

**Reviewer 2:**

Comments: The authors use more than two years of CATS data to examine the diurnal cycles of the aerosol loading on global scale. Their results show that a strong peak at 6 am local time in aerosol extinction profile over North Africa during the June-November season. This finding is exciting and brand new. I would recommend this manuscript be published in ACP after a few minor changes

Response: We thank the reviewer for his/her encouragement and his/her thoughtful comments

Comments: (1) In Figure 5, there are some spikes above 2 km in the aerosol extinction vertical profiles seen in the CATS data, but not present in the CALOP data. Are they due to the cloud screening differences between CATS and CALIOP?

*Response:* We suspect that the high spikes were introduced by a bug in the code which allowed a very small number of larger extinction values through. This has been fixed, and the spikes are no longer present. The overall shapes of the profiles remain unchanged.

Comments: (2) Line# 353-354, unlike CALIOP, MODIS Aqua aerosol products are only available in the early afternoon, but not in the early morning, since the algorithm only performs retrieval over daytime.

Response: We have revised the sentence to "as CALIOP provides early morning and afternoon over passes, and Aqua MODIS has an over pass time after local noon,"

Comments: (3) Line# 355-356, please add a sentence or two to briefly elaborate what aerosol above cloud issues are as reported by Rajapashe et al., (2017).

*Response:* This study has been explained in Section 2.2. To avoid duplication, we have revised the sentence to "It is also possibly due to aerosol above cloud related issues as reported by Rajapakshe et al. (2017), as explained in Section 2.2"

Comments: (4) Line# 358, please spell out "AGL".

Response: Done. We have added "Above Ground Level (AGL)"

Comments: (5) The aerosol extinction at 1064 nm may not be as sensitive to the fine mode aerosols (such as smoke and urban pollutant aerosols) compared to the coarse mode aerosols (such as dust). The authors probably should add a few sentences to address this

Response: Great point. We have added the following discussions to address this issue. "Still, readers shall be aware that AOD retrievals at the 1064 nm are less sensitive to fine mode aerosols such as smoke and pollutant aerosols, compared to coarse mode aerosols such as dust aerosols. Thus, an investigation of diurnal variations of aerosol properties at the visible channel may be also needed for a future study."

**Short Comments by Mark Vaughan and Stuart Young**

Comments: This paper compares the aerosol extinction profiles and aerosol optical depths (AODs) retrieved by the CATS lidar with similar quantities retrieved by AERONET, MODIS, and CALIOP. To our knowledge, this is the first ever in-depth assessment of satellite-derived AODs measured/retrieved in the near-infrared, and thus should be of great interest to several different groups in the aerosol research community. Overall, the authors have done a good job in bringing multiple analyses together. However, we find several places where additional analyses are warranted and where more in-depth discussions will help strengthen the final manuscript.

*Response: We thank Mark Vaughan and Stuart Young and we appreciate the suggestions and comments which we believe are shaping this paper into a better paper*

**General Remarks**

Comments: When filtering the extinction coefficients retrieved from the CATS and CALIOP measurements, the authors say that candidate extinction coefficients were constrained to a "nominal range" of 0 to 1.25 km–1, and that "all near zero negative values" are set to zero (page 6, lines 114–119). Presumably these "near zero negative values" that were set to zero were not actually removed from consideration, but instead were included in subsequent data averaging operations (the writing in this section is not sufficiently clear on this point). Changing negative values to zeros prior to averaging is not statistically valid, as it guarantees high biases in the estimated mean values. Reporting these high biases erroneously improves the comparisons of lidar-derived optical depths with those obtained by other sensors. To avoid this, all CATS and CALIOP mean values should be correctly recalculated before the final version of this manuscript is published.

Response: This is an excellent point. First, when calculating AOD and AOD climatology, we used the CATS and CALIOP derived AOD values. Thus, this is no need for us to detailing with negative extinction coefficients and we have revised the paper to reflect the issue. We did, however, apply the constraint that AOD values which came from retrievals containing extinction coefficients greater than 1.25 km–1 be excluded to avoid using AOD values from what are likely cloud contaminated profiles. We have added the following discussions.

"In AOD related studies, CAT and CALIOP reported AOD values are used. However, although not derived in this study, only AOD values with corresponding aerosol vertical extinction that meet the QA criteria as mentioned in Sections 2.1 and 2.2 were used."

Still, the extinction coefficients are used in estimating aerosol vertical distributions. As suggested we have revised our calculations and included those negative values, instead of setting them to zero. Note that similar approaches have been adopted for passive-based AOD studies, where

negative AODs are used to reduce high bias in long term studies (Remer et al., 2005). We have made the changes in the text accordingly.

"Extinction was also constrained using a threshold as provided in the CATS data catalog (Extincton\_Coefficient\_1064\_Fore\_FOV <= 1.25 km-1), similar to several previous studies (Redemann et al., 2012; Toth et al., 2016). Only profiles with extinction coefficient values less than 1.25 km-1 are included in this study. Small negative extinction coefficient values, however, are included in aerosol profile related analysis, to reduce potential high biases in computed mean profiles. Note that a similar approach has also be conducted in deriving passive-based AOD climatology (e.g. Remer et al., 2005)."

Remer, L.A., Y.J. Kaufman, D. Tanré, S. Mattoo, D.A. Chu, J.V. Martins, R. Li, C. Ichoku, R.C. Levy, R.G. Kleidman, T.F. Eck, E. Vermote, and B.N. Holben, 2005: The MODIS Aerosol Algorithm, Products, and Validation. J. Atmos. Sci., **62**, 947–973, https://doi.org/10.1175/JAS3385.1

Comment: While the main body of the text emphasizes the correlations between the CATS retrievals and the other data sets (e.g., lines 186–208), the authors do not provide any quantitative statements about the magnitudes of the CATS AODs or the differences between the different AOD estimates. Given that this paper is (to our knowledge) the first ever in-depth look at 1064 nm AOD, tables showing global and regional mean values and quantifying the CATS AOD estimates relative to the other sensors would add significantly to the value delivered by this paper. Profiles of the relative CATS-CALIOP extinction coefficient differences (i.e., (CATS(z) - CALIOP(z)) / CALIOP(z)) would be especially interesting.

*Response:* We have added a table to include regional and global means. Still, we documented that the differences may also be introduced by sampling differences of the sensors.

| Region         | Latitude        | Longitude        | Mean CATS AOD
(DJFMAM/JJASON) | Mean CALIOP AOD
(DJFMAM/JJASON) |
|----------------|-----------------|------------------|----------------------------------|------------------------------------|
| Global         | 52°S-52°N       | 180°W-180°E      | 0.09/0.10                        | 0.09/0.09                          |
| India          | 7.5°N - 32.5°N  | 65°E - 85°E      | 0.22/0.26                        | 0.22 /0.28                         |
| Africa - North | 2.5°N - 22.5°N  | 35°W - 20°E      | 0.26/0.23                        | 0.30 /0.25                         |
| Africa - South | 17.5°S - 2.5°N  | 0° - 30°E        | 0.14/0.22                        | 0.15 /0.13                         |
| Middle East    | 12.5°N - 27.5°N | 35°E - 50°E      | 0.22/0.33                        | 0.26/0.35                          |
| China          | 27.5°N - 37.5°N | 110°E -
120°E | 0.19/0.18                        | 0.21 /0.16                         |

"Table 2. CALIOP and CATS mean aerosol optical depth for regions as highlighted in Figure 6 and globally between +/- 52° latitude."

We have also added a plot of the difference (CATS(z)-CALIOP(z)) in Appendix A. As CALIOP extinction values become very small, the ratio of (CATS(z)-CALIOP(z))/CALIOP(z) has a tendency to grow very large from just a few data points and greatly impacts the standard deviation. Thus we plotted only the difference and did not include (CATS(z)-CALIOP(z))/CALIOP(z) and error bars with this particular plot.

Comment: In section 3.1.1., CATS observations are compared with other observations made within  $\pm 30$  mins and  $\pm 0.4$  degrees. For aerosols, this is probably not too much of a problem a lot of the time, but we have seen numerous cases where there can be large differences in the scenes being observed (e.g., see Omar et al., 2013: "In 45% of the coincident instances CALIOP and AERONET do not agree on the cloudiness of the scenes."). For AERONET, the comparisons may be improved by imposing another criterion, i.e., that the AERONET AODs made at the closest times preceding and following the CATS observations not vary by more than x%. A similar filter for potential spatial differences could include wind speed and direction (e.g., Lopes et al., 2013) and the spatial separations of the AERONET sites and the CATS observations. (This is likely to be quite a bit messier.)

Response: We have included the references as suggested and reminded readers that the collocation criteria may have impacts to the results due to the spatial and temporal sampling methods chosen.

"Note that as suggested by Omar et al., 2013, the choices of spatial and temporal collocation windows have an effect on collocation results. However, we consider this as a topic beyond the scope of this study"

Comment: While the authors point out a number of differences between the CATS retrievals and those derived from other sensors, they typically do not attempt to identify the causes of these differences. For example, based on the scaling factors in the linear regressions, the CATS AODs are lower than all of the AODs with which they are being compared (i.e., AERONET in Figure 1, MODIS in Figure 2, and CALIOP in Figure 3). This is perhaps not surprising for the AERONET and MODIS comparisons, but the cause for the CATS-CALIOP differences is not as obvious. Differences between CALIOP and MODIS at visible wavelengths are frequently explained by CALIOP's low daytime detection sensitivity and the missed detection of some of the vertical extent of the aerosol layer (e.g., Kim et al., 2017 and Toth et al., 2018). Can the authors enumerate the possible causes that would explain the disparities between CATS and CALIOP?

Response: Slopes in linear regressions can often be biased by outliers. In Figure 6, which are spatial plots of AODs from CALIOP and CATS, differences are less noticeable for the DJFMAM season. For the JJASON season, CATS AODs are lower at certain regions (Middle East, India, and North Africa) and higher over other regions (South Africa). The cause of those discrepancies, however, is unclear to us. Also, Version 2 of the CATS data are used in this study, and we expect some difference with the version 3 of CATS data. To really explore the issue, it deserves a paper of its own. Thus, we leave this topic to a future paper.

Comment: Furthermore, given the lower CATS AODs shown in Figure 2, it's surprising to see that the CATS extinctions coefficients shown in Figure 5 are typically larger than CALIOP at all altitudes, and that the closest agreement is over land (where CATS slightly underestimates CALIOP at lower altitudes). Again, some discussion of the possible causes of this paradox would be welcome.

Response: First, there is a call from the community to avoid using slopes from the regression analysis as they are prone to noisy data, and we are kind of agree with them. Statistically, we expect a high percentage of small AODs versus large AODs. Still, slopes are dominated by high AOD cases, while the averaged profiles may be more dominated by low AOD cases. This could explain the difference.

Comment: The results shown in Figure 5 are a prime candidate for further investigation into the underlying causes of the differences. Except for the over land case, CATS extinction profiles consistently and significantly overestimate CALIOP extinction profiles. It seems that there are four likely suspects in causing this (always keeping in mind that that all four could be collaborating in various nefarious ways to bring this about): layer detection, cloud-aerosol discrimination (including inadequate boundary layer cloud clearing), lidar ratio selection, and calibration. Of these four, the easiest to investigate (at least at a superficial level) is lidar ratio. The table below shows the default lidar ratios assigned by each instrument.

|                      | , the default fidul f | and assigned by ca |
|----------------------|-----------------------|--------------------|
| Aerosol Type         | CATS                  | CALIOP             |
| Dust                 | 40 sr                 | 44 sr              |
| Dust mixture (a)     | 40 sr                 | N/A                |
| Polluted dust (a)    | N/A                   | 48 sr              |
| Dusty marine (a)     | N/A                   | 37 sr              |
| Marine               | 25 sr                 | 23 sr              |
| Clean/background     | 35 sr                 | 30 sr              |
| Polluted continental | 35 sr                 | 30 sr              |
| Smoke                | 40 sr                 | 30 sr              |
| Volcanic (b)         | 35 sr                 | 44 sr              |

a) CATS identified dust mixtures over land and water; CALIOP identifies 'polluted dust' over land only and 'dusty marine' over water only.

(b) For CATS, all aerosol above 10 km is classified as volcanic. For CALIOP, volcanic aerosol is identified in the stratosphere only.

Since the CATS marine lidar ratio is large than the CALIOP marine lidar ratio, and the CATS dust mixture lidar ratio is larger than the CALIOP dusty marine lidar ratio (and CATS smoke and polluted continental lidar ratios are greater than their CALIOP counterparts as well), then, all other things being equal, one should expect the CATS over-ocean extinction profiles to be uniformly larger than the CALIOP extinction profiles. (But are all other things actually equal?)

The case is less clear over land. But since the CATS dust lidar ratio is less than the CALIOP dust lidar ratio and the CATS dust mixture lidar ratio is less than the CALIOP polluted dust lidar

ratio, if we assume that the over-land aerosols detected in this study are dominated by dust (which might not be a bad assumption?), then perhaps the over-land profile comparison makes sense too. (All other things being equal, that is...)

**Response:**

We have added a discussion of potential sources of CATS-CALIOP extinction and AOD differences in the text:

"In addition, due to the precessing orbit of the ISS, the CATS sampling is irregular and very different compared to the sun-synchronous orbits of the A-Train sensors. These orbital differences between CATS and CALIOP make comparing the data from these two sensors challenging since they are fundamentally observing different locations of the Earth at different times. Thus, we shouldn't expect the extinction profiles and AOD from these two sensors to completely agree. Additionally, there are other algorithm and instrument differences that can lead to differences in extinction coefficients and AOD. Over land where dust is the dominant aerosol type, differences in lidar ratios between the two retrieval algorithms (CATS uses 40 sr while CALIOP uses 44 sr), can cause CATS extinction coefficients that are up to 10% lower than CALIOP, potentially explaining the higher CALIOP extinction values in Figure 5e. Over ocean, especially during daytime, differences in CATS and CALIOP lidar ratios for marine and smoke aerosols, as well as issues with CATS cloud-aerosol discrimination in V2-01 for daytime observations, can cause CATS extinction coefficients that are as much as 25% higher than CALIOP (Figure 5b and 5d). Yorks et al. (2019) shows examples of these daytime cloud-aerosol discrimination issues in V2-01 data, which have been improved for CATS V3-00 data. A brief analysis using 3 months of CATS V3-00 data showed improvement in agreement for AOD, but some differences were still evident in the extinction vertical profiles. These remaining differences, as well as the differences observed in nighttime only profiles (Figure 5c) are likely attributed to differences in CATS and CALIOP 1064 nm backscatter calibration. Pauly et al. (2019) reports that CATS attenuated total backscatter is about 18% higher than CALIOP due to calibration uncertainties for both sensors."

Comment: The CATS extinction profiles shown in Figures 5 and 10 peak at altitudes some hundreds of meters higher than do CALIOP's, except over land. While CALIOP's profiles show almost no roll off until about the last range bin above the surface, the CATS profiles start dropping off below about 500 m, or at approximately 8 to 9 range bins above the surface. What is happening here? Is CATS altitude registration and/or surface detection the culprit? Or is the cloud filter too aggressive in the boundary layer (i.e., are strongly scattering aerosols being misclassified as clouds)? Irrespective of the underlying cause(s), is this behavior a major source of AOD differences between CATS and CALIOP?

Response: The 2 biggest issues in the CATS V2-01 data were the daytime calibration and the daytime cloud-aerosol discrimination. A CATS paper in preparation (Yorks et al., 2019) has included details about the cloud-aerosol discrimination issues, while Rebecca Pauly's 1064 nm calibration paper has a lot of details about the new daytime calibration. We have checked this

issue by reprocessing the analysis using 3 months of V3 data and we found an improvement in agreement for AOD, but with some differences still evident in the vertical profiles.

Comment: The seasonal maps (Figure 6) show that the CALIOP AODs exceed those of CATS over the Arabian Peninsula, and to a smaller degree over the African region bordering the Gulf of Guinea. Can this also be explained by differences in lidar ratio selection, or are there other factors at work?

Response: We suspect the difference in retrieval method as mentioned above may contribute. Also, CALIOP provides early morning and afternoon overpasses while CATS can observe at near all solar hours, the differences may also be associated with these sampling differences.

**Specific Comments**

Comment: page 4, line 85: provide a reference for "Feature Type Score"

Response: We have added the reference to the text.

Comment: page 5, line 107: did the authors also consider potential sources of bias errors; e.g., unusually large or small calibration coefficients, or large values of overlying integrated attenuated backscatter?

*Response:* We have adopted the QA steps from a few previous papers such as Campbell et al., 2012; Toth et al., 2016; 2018. The thresholds for the above mentioned criteria are not mentioned and used in those previous papers, and thus we didn't include the check as suggested.

Comment: page 5, line 113: "Extinction\_Coefficient\_Uncertainty\_1064\_Fore\_FOV  $\leq$  10 km-1"; despite the heritage from Campbell et al. (2012), using relative uncertainties still makes much, much more sense. Given the noise in the CATS daytime measurements, an uncertainty threshold of 10 km-1 might be reasonable for an estimated extinction coefficient of 1 km-1. However, for the substantially smaller extinction coefficients (e.g., 0.01 km-1 to 0.1 km-1) that make up a very large majority of the measurements, an uncertainty threshold of 10 km-1 seems prohibitively large.

Response: Agreed. Since we have to apply the thresholds to all observations, lowering the threshold may exclude heavy plumes that may indeed be valid. Also, other QA steps, along with this threshold are also used, as thus, we expect some of the issues as mentioned can be captured by other QA steps. Thus, the QA steps remain unchanged.

Comment: page 6, line 128: distinguish between laser spot size (~70 m) and receiver footprint diameter at the Earth's surface (~90 m).

Response: We have changed the sentence to "with a laser spot size of around 70 m"

Comment: page 6, line 129: say which version of the CALIPSO data products was used (version 4.1, right?)

Response: We have included "CALIOP Level 2.0 Version 4.1" in the sentence.

Comment: page 7, line 137: "signal-to-noise", not "single to noise"

Response: Done.

Comment: page 7, line 148: "Atmospheric\_Volume\_Description = 3 (aerosol only)"; note that in the CALIPSO version 4.1 data products, 3 indicates tropospheric aerosols and 4 indicates stratospheric aerosols. Were stratospheric aerosols excluded accidentally or deliberately? (Previous versions of the CALIPSO data products did not differentiate between tropospheric and stratospheric aerosols. In these earlier products, requiring the atmospheric volume description to equal 3 would correctly identify all aerosol data.) If accidentally, please correct the calculations. If deliberately, please explain why.

*Response:* We have updated this to include Atmospheric\_Volume\_Description = 4 as well, and updated the text accordingly.

"Atmospheric Volume Description = 3 or 4 (aerosol only)"

Comment: page 8, line 163: logarithmic interpolation, correct? Also, please state the actual value of the Ångström exponent given by Shi et al.

Response: Yes. The Angstrom exponent value is computed for each AOD retrieval. We have revised the discussion to avoid confusion. "Here we assume the angstrom exponent value, computed using instantaneous AOD retrievals at the 860 and 1240 nm, remains the same for the 860 to 1064 nm wavelength range, similar to what has been suggested by Shi et al., (2011; 2013)."

Comment: page 8, line 170: while "AERONET data are considered as the ground truth for evaluating CATS retrievals", it should be noted that there are very few AERONET sites in remote oceans. Do MODIS retrievals substitute as the gold standard in these places?

Response: Even though a better performance can be expected from MODIS aerosol retrievals over ocean versus over land, we still think that only AERONET data should be used for ground truth, as instantaneous retrievals from passive sensors suffer from various issues such as cloud contamination.

Comment: page 9, line 186–187: some discussion on the rationale for the choices of  $\pm 0.4^{\circ}$  and  $\pm 30$  minutes would be helpful in evaluating the strength of the comparisons.

Reponses: We picked this threshold following a few previous papers (e.g. Toth et al., 2018). We have added discussions in the text to further clear this issue:

"Note that as suggested by Omar et al., 2013, the choices of spatial and temporal collocation windows have an effect on collocation results. However, we consider this as a topic beyond the scope of this study"

Comment: page 9, line 193: how frequently do "profiles with all retrieval fill values" occur in the CATS data set?

*Response: We have examined the dataset and found that profiles in which there were no cloud or aerosol made up about 5.4% (3583933/65792363) of all profiles. The text has been updated accordingly.*

"Such profiles containing all retrieval fill values were found to make up approximately 5.4% of all CATS profiles in the dataset."

Comment: page 9, line 194: as a rule of thumb, how close to sunrise and sunset can reliable AERONET measurements be obtained?

*Response:* We are not aware if any study have been conducted on this issue. Because it is hard to "validate" AERONET observations. But it is an interesting topic for a future paper.

Comment: page 11, line 244: The authors say, "using over land (ocean) daytime data only, for a total of 171 (1207) collocated pairs." Here we echo the remarks of an anonymous reviewer commenting on a paper for which one of us (Mark Vaughan) is a coauthor (see https://doi.org/10.5194/acp-2018-1090-RC1).

Way back in 2010 Prof. Robock pleaded with us to end this misuse of parentheses [Robock, A. (2010), Parentheses are (are not) for references and clarification (saving space), Eos Trans. AGU, 91(45), 419–419, doi:10.1029/2010EO450004]. My understanding is that one of the publishers in our field has specifically written it out of their style guide. I read pretty widely and the only genre of writing where I have experienced this application of parentheses is in the atmospheric sciences journals. I hope the authors will consider rewriting this sentence.

Response: Done. We have rewritten the sentence.

Comment: page 11, line 245: The authors say, "daytime data from both CALIOP and CATS are expected to be nosier due to solar contamination". While this is true, the day-night differences at 1064 nm are very different for the two lidars. CATS daytime SNR is substantially worse than CATS nighttime SNR, whereas CALIOP daytime SNR is only marginally worse than CALIOP nighttime SNR. The primary reason for this is that CALIOP 1064 nm detector is an avalanche photodiode for which the dark counts contribute substantial amounts of noise irrespective of the external lighting conditions. Moreover, while CATS 1064 nm nighttime SNR is much higher than CALIOP 1064 nm nighttime SNR, for daytime measurements the CALIOP SNR is higher. This should be explained in greater detail in a forthcoming CATS calibration paper.

*Response:* Great comment. But we think those comments should be included in a future paper, hopefully written by one of the coauthors.

Comment: page 12, line 260: "it is speculated". Who's doing this speculating? If it's the authors, then come right out and say so!

Response: We have revised the sentence to "although we speculate"

Comment: page 14, line 311: The authors say, "the shapes of the CATS and the CALIOP nm extinction vertical profile are very similar for all three cases". This qualitative assessment would be much more meaningful if it was augmented by a set of quantitative metrics (e.g., profiles of (CATS(z) - CALIOP(z)) / CALIOP(z), with error bars to indicate the magnitude of the variability in the ratios).

Response: We have included a plot of CATS(z) - CALIOP(z) (Appendix A) for the mean CATS and CALIOP vertical profiles. As CALIOP extinction values become very small, the ratio of (CATS(z)-CALIOP(z))/CALIOP(z) has a tendency to grow very large from just a few data points and greatly impacts the standard deviation. Thus we plotted only the difference and did not include (CATS(z)-CALIOP(z))/CALIOP(z) with error bars with this particular plot.

Comment: page 18, line 405: The authors say, "nighttime retrievals from CATS *are considered to be less noisy* than daytime" (emphasis added). This sentence suggests that there might be some debate about day versus night noise magnitudes. There is no such debate. The fact is that "nighttime retrievals from CATS *are significantly and demonstrably less noisy* than daytime retrievals".

**Response: We have used the wording as suggested. Thanks for the comment.**

Comment: page 23, lines 514–517: The authors' conclusions reinforce the conventional wisdom. However, we think it's important to emphasize that at present these conclusions are highly tentative, and will remain so until a comprehensive analysis of the CATS calibration accuracy and stability is completed.

Response: We have added the comment as suggested:

[revised manuscript text omitted]
, as well as issues with CATS cloud-aerosol discrimination in V2-01 for daytime 363 observations, can cause CATS extinction coefficients that are as much as 25% higher than 364 CALIOP (Figure 5b and 5d). Yorks et al. (2019) shows examples of these daytime cloud-aerosol 365 discrimination issues in V2-01 data, which have been improved for CATS V3-00 data. A brief 366 analysis using 3 months of CATS V3-00 data showed improvement in agreement for AOD, but 367 some differences were still evident in the extinction vertical profiles. These remaining differences, as well as the differences observed in nighttime only profiles (Figure 5c) are likely 368 369 
[revised manuscript text omitted]
| 700 | https://doi.org/10.1175/JAS3385.1, 2005.                                                      | Formatted: Font: Not Italic, Not Highlight                                            |
|     |                                                                                               |                                                                                       |

| 702 | Ryder, C. L., McQuaid, J. B., Flamant, C., Rosenberg, P. D., Washington, R., Brindley, H.     |
|-----|-----------------------------------------------------------------------------------------------|
| 703 | E., Highwood, E. J., Marsham, J. H., Parker, D. J., Todd, M. C., Banks, J. R., Brooke, J.     |
| 704 | K., Engelstaedter, S., Estelles, V., Formenti, P., Garcia-Carreras, L., Kocha, C., Marenco,   |
| 705 | F., Sodemann, H., Allen, C. J. T., Bourdon, A., Bart, M., Cavazos-Guerra, C.,                 |
| 706 | Chevaillier, S., Crosier, J., Darbyshire, E., Dean, A. R., Dorsey, J. R., Kent, J.,           |
| 707 | O'Sullivan, D., Schepanski, K., Szpek, K., Trembath, J., and Woolley, A.: Advances in         |
| 708 | understanding mineral dust and boundary layer processes over the Sahara from Fennec           |
| 709 | aircraft observations, Atmos. Chem. Phys., 15, 8479-8520, https://doi.org/10.5194/acp-        |
| 710 | 15-8479-2015, 2015.                                                                           |
| 711 | Shi Y., Zhang, J., Reid, J. S., Hyer, E., and Hsu, N. C.: Critical evaluation of the MODIS    |
| 712 | Deep Blue aerosol optical depth product for data assimilation over North Africa, Atmos.       |
| 713 | Meas. Tech., 6, 949-969, doi:10.5194/amt-6-949-2013, 2013.                                    |
| 714 | Shi Y., Zhang J., Reid J. S., Hyer E. J., Eck T. F., and Holben B. N.: A critical examination |
| 715 | of spatial biases between MODIS and MISR aerosol products - application for potential         |
| 716 | AERONET deployment, Atmos. Meas. Tech., 4, 2823–2836, 2011.                                   |
| 717 | Stephens, G. L., and coauthors: The CLOUDSAT mission and the A-TRAIN, Bulletin of the         |
| 718 | American Meteorological Society, 83(12), 1771-1790. https://doi.org/10.1175/BAMS-             |
| 719 | 83-12-1771 , 2002.                                                                     |
| 720 | Tiwari, S., Srivastava, A. K., Bisht, D. S., Parmita, P., Srivastava, M. K., and Atri, S. D.: |
| 721 | Diurnal and seasonal variaition of black carbon and PM2.5 over New Delhi, india:              |
| 722 | Influence of meteorology, Atmos. Res, 125, 50-62, doi:10.1016/j.atmos.res.2013.01.011,        |
| 723 | 2013.                                                                                         |

| 724 | Toth, T. D., Campbe  | ll, J. R., Reid, J. S., | Tackett, J. L., Vaug   | ghan, M. A., 2  | Zhang, J., &  |
|-----|----------------------|-------------------------|------------------------|-----------------|---------------|
| 725 | Marquis, J. W.:      | Minimum aerosol lay     | yer detection sensitiv | ities and their | r subsequent  |
| 726 | impacts on aeros     | ol optical thickness    | retrievals in CALIPS   | SO level 2 da   | ata products. |
| 727 | Atmospheric          | Measurement             | Techniques,            | 11(1),          | 499–514.      |
| 728 | https://doi.org/http | os://doi.org/10.5194/ar | nt-11-499-2018, 2018   |                 |               |

- 729 Toth, T. D., Zhang, J., Campbell, J. R., Reid, J. S., & Vaughan, M. A.: Temporal variability
- of aerosol optical thickness vertical distribution observed from CALIOP, Journal of
  Geophysical Research: Atmospheres, 121(15), 9117–9139.
  https://doi.org/10.1002/2015JD024668, 2016.
- Vaughan, M., Garnier, A., Josset, D., Avery, M., Lee, K.-P., Liu, Z., Hunt, W., Pelon, J.,
  Tackett, J., Getzewich, B., Kar, J., and Burton, S.: CALIPSO Lidar Calibration at 1064
  nm: Version 4 Algorithm, in preparation, 2018.
- Wang, J., Liu, X., Christopher, S. A., Reid, J. S., Reid, E. A., and Maring, H.: The effects of
  non-sphericity on geostationary satellite retrievals of dust aerosols, Geophys. Res. Lett.,
  30(24), 2293, doi:10.1029/2003GL018697, 2003.
- Winker, D. M., and coauthors: Overview of the CALIPSO Mission and CALIOP Data
  Processing Algorithms. Journal of Atmospheric and Oceanic Technology, 26(11), 2310–
  2323. https://doi.org/10.1175/2009JTECHA1281.1, 2009.
- Yorks, J. E., McGill, M. J., Palm, S. P., Hlavka, D. L., Selmer, P. A., Nowottnick, E.,
  Vaughan, M. A., Rodier, S., and Hart W. D.: An Overview of the CATS Level 1 Data
  Products and Processing Algorithms, Geophys. Res. Let., 43,
  doi:10.1002/2016GL068006, 2016.

Deleted: Yorks, J. E., Palm, S. P., Hlvaka, D. L., McGill, M. J., Nowotnick, E., Selmer, P. A., Hart, W. D.: The Cloud-Aerosol Transport System (CATS)¶ algorithm theoretical basis document. Available at http://cats.gsfc.nasa.gov/media/docs/CATS\_ATBD.pdf, 2015.¶

| 751 | Yorks, J. E., Rodier, S.D., Nowottnick, E., Selmer, P.A., McGill, M.J., Palm, S.P., and   |
|-----|-------------------------------------------------------------------------------------------|
| 752 | Vaughan, M. A.: CATS Level 2 Vertical Feature Mask Algorithms and Data Products:          |
| 753 | An Overview and Initial Assessment, Atmos. Meas. Tech. Discuss., in preparation.          |
| 754 | Yoshida M., Kikuchi, M., Nagao, T. M., Murakami, H., Nomaki, T., and Higurashi, A.:       |
| 755 | Common Retrieval of Aerosol Properties for Imaging Satellite Sensors, Journal of the      |
| 756 | Meteorological Society of Japan. Ser. II, Article ID 2018-039, [Advance publication],     |
| 757 | https://doi.org/10.2151/jmsj.2018-039, 2018.                                              |
| 758 | Zhao, X. J., Zhang, X. L., Xu, X. F., Xu, J., Meng, W., and Pu, WW.: Seasonal and diurnal |
| 759 | variation of ambient PM2.5 concentrations in urban and rural environments in Beijing,     |
| 760 | Atmos. Environ., 43, 2893-2900, doi: 10.106/j.atmosenv.2009.03.009., 2009.                |

| 762 | Table 1. Geographic ranges, height above ground level of maximum extinction, diurnal          |
|-----|-----------------------------------------------------------------------------------------------|
| 763 | extinction range at height of maximum extinction, and time (local) of peak extinction for the |

boxed red regions in Figure 6 and vertical profiles shown in Figures 12 and 13.

| DJFMAM/JJASON  |                 |                  |                                         |                                                                             |                                                                      |  |
|----------------|-----------------|------------------|-----------------------------------------|-----------------------------------------------------------------------------|----------------------------------------------------------------------|--|
| Region         | Latitude        | Longitude        | Height AGL
(m) of Max.
Extinction | Extinction Range
(km .1 ) at Height AGL of Max.
Extinction | Time of Peak
Extinction at
Height AGL of
Max.
Extinction |  |
| India          | 7.5°N - 32.5°N  | 65°E - 85°E      | 180/240                                 | 0.109-0.131/0.138-0.182                                                     | 6 am/6 am                                                            |  |
| Africa - North | 2.5°N - 22.5°N  | 35°W - 20°E      | 420/480                                 | 0.107-0.130/0.098-0.121                                                     | 12 pm/6 am                                                           |  |
| Africa - South | 17.5°S - 2.5°N  | 0° - 30°E        | /420                                    | /0.090-0.100                                                                | /6 am                                                                |  |
| Middle East    | 12.5°N - 27.5°N | 35°E - 50°E      | 240/180                                 | 0.093-0.116/0.081-0.135                                                     | 6 am/0 am                                                            |  |
| China          | 27.5°N - 37.5°N | 110°Е -
120°Е | 240/240                                 | 0.107-0.154/0.085-0.133                                                     | 6 am/6 am                                                            |  |

| 766        |                                     |                                        |                                |                                         |                                           |                                                                                                |
|------------|-------------------------------------|----------------------------------------|--------------------------------|-----------------------------------------|-------------------------------------------|------------------------------------------------------------------------------------------------|
| 767        |                                     |                                        |                                |                                         |                                           |                                                                                                |
| 768
769 | Table 2. CALIO
and globally bety | P and CATS mean
ween +/- 52° latitu | n aerosol optica
de.        | l depth for regions as                  | s highlighted in Figure                   | 6 • Formatted: Font: 12 pt, Not Italic, Font color: Auto
|            | Region                              | Latitude                               | Longitude               | Mean CATS AOD
(DJFMAM/JJASON) | Mean CALIOP AOD
(DJFMAM/JJASON) | Formatted: Font: 12 pt, Not Italic, Font color: Auto Formatted Table                           |
|            | Global                       | 52°S-52°N                       | 180°W-180°E             | 0.09/0.10                               | 0.09/0.09                                 |                                                                                                |
|            | India                        | 7.5°N - 32.5°N                  | 65°E - 85°E             | 0.22/0.26                               | 0.22/0.28                                 |                                                                                                |
|            | Africa - North               | 2.5°N - 22.5°N                  | 35°W - 20°E             | 0.26/0.23                               | 0.30 /0.25                                |                                                                                                |
|            | Africa - South               | 17.5°S - 2.5°N                  | 0° - 30°E               | 0.14/0.22                               | 0.15 /0.13                                |                                                                                                |
|            | Middle East                         | 12.5°N - 27.5°N                 | 35°E - 50°E             | 0.22/0.33                               | 0.26/0.35                          |                                                                                                |
|            | China                        | 27.5°N - 37.5°N                 | 110°E -
120°E | 0.19/0.18                               | 0.21/0.16                                 |                                                                                                |
| 770        |                                     |                                        |                                |                                         |                                           | Formatted: Line spacing: Multiple 1.08 li                                                      |

**771 Figure Captions**

772

Figure 1. Collocated AERONET 1020 nm AOT vs. CATS 1064 nm AOD a) without CATS QA
applied, and b) with CATS QA applied.

Figure 2. Collocated MODIS C6.1 a) Terra and b) Aqua estimated 1064 nm AOD vs. CATS
1064 nm AOD with CATS QA applied.

Figure 3. Collocated CALIOP 1064 nm AOD vs. CATS 1064 nm AOD with CATS QA applied
for a) both day and night, b) nighttime over-land, c) nighttime over-water, d) daytime over-land,
e) daytime over-water.

780 Figure 4: CATS 1064 nm AOD a) as a function of local time for the globe, and b) as a function

781 of local time for areas south of -25 degrees. The difference between CATS 1064 nm AOD and

782 AERONET 1020 nm AOD as a function of local time is shown in c). The mean is represented

783 by the blue line, while the median is the green line.

Figure 5. CATS and CALIOP vertical profiles of 1064 nm extinction for a) all profiles, b)
daytime only, c) nighttime only, d) over-water, and e) over land.

**Figure 6.** Mean AOD (1064 nm) by season for a) DJFMAM CATS, b) JJASON CATS, c)

DJFMAM CALIOP, d) JJASON CALIOP, e) DJFMAM MODIS Aqua, and f) JJASON MODIS
 Aqua. Red boxes indicate locations of regional vertical distributions in Figures 12 and 13.

**Figure 7.** Mean CATS AOD (1064 nm) by season for a) DJFMAM below 1 km AGL, b)

790 JJASON below 1 km AGL, c) DJFMAM 1-2 km AGL, d) JJASON 1-2 km AGL, e) DJFMAM

above 2 km AGL, and f) JJASON above 2 km AGL.

Figure 8. Seasonal Mean AOD (1064 nm) binned by every 6-hours for a) DJFMAM 0 UTC, b)
JJASON 0 UTC, c) DJFMAM 6 UTC, d) JJASON 6 UTC, e) DJFMAM 12 UTC, f) JJASON 12
UTC, g) DJFMAM 18 UTC, and h) JJASON 18 UTC.

Figure 9. Maximum minus minimum mean seasonal AOD (1064 nm) for a) DJFMAM, and b)JJASON.

Figure 10. Global mean 6-hourly vertical profiles of CATS 1064 nm extinction for a) DJFMAM
all profiles, b) DJFMAM water profiles, c) DJFMAM not-water profiles, e) JJASON all profiles,
f) JJASON water profiles, g) JJASON not-water profiles.

Figure 11. Global mean 6-hourly local time (0:00 am, 6:00 am, 12:00 pm and 6:00 pm) vertical
profiles of CATS 1064 nm extinction for a) DJFMAM all profiles, b) DJFMAM water profiles,
c) DJFMAM not-water profiles, d) JJASON all profiles, e) JJASON water profiles, f) JJASON
not-water profiles.

**Figure 12.** DJFMAM 6-hourly average (local time; 0:00 am, 6:00 am, 12:00 pm and 6:00 pm)

vertical profiles of CATS 1064 nm for locations shown in Figure 6a; a) Africa-north, b) Middle
East, c) India, and d) Northeast China.

- **Figure 13.** JJASON 6-hourly average (local time; 0:00 am, 6:00 am, 12:00 pm and 6:00 pm) vertical profiles of CATS 1064 nm for locations shown in Figure 6b; a) Africa-north, b) Africa-809
- 810 south, c) Middle East, d) India, and e) Northeast China,

811

---

## Referee Report (RR1)

I think that the authors have adequately addressed the comments made by the reviewers in the revised version of the manuscript. Therefore, I have no further comments.

---

## Referee Report (RR2)

**Review of "Investigation of CATS aerosol products and application toward global diurnal variation of aerosols" by Logan Lee, Jianglong Zhang, Jeffrey S. Reid, and John E. Yorks**

reviewed by Mark Vaughan

This paper compares the spatial and temporal distributions of the aerosol optical depths retrieved at 1064 nm by the CATS lidar aboard the International Space Station to the optical depths measured by AERONET (at 1020 nm) and the optical depths retrieved by MODIS and CALIOP (at 1064 nm).

This is the second version of this manuscript that I have read, but the first for which I've been asked to provide an invited (as opposed to contributed) review.

My primary comment about this second version is that the authors' do not provide enough information for readers to confidently assess the quality of the CATS AOD estimates relative to those provided by the other sensors. In particular, relying on correlation coefficients alone to characterize the comparisons is insufficient. Consider the two series defined by $y_2 = 2\,x$ and $y_4 = 4\,x$. While $y_2$ and $y_4$ are perfectly correlated – i.e., they have a correlation coefficient of 1 – in the mean, $y_4$ is twice as large as $y_2$. So in addition to the correlation coefficients they already provide, the authors should also provide means and standard deviations for each of the datasets being compared. While Table 2 is a fine start, more is needed.

In a similar vein, regarding figures 1 through 3, black-on-black overplotting of data points in high data density regions reduces the information content of the figures. So, in addition to the figures, the authors should also cite the descriptive statistics (e.g., min, max, median, mean, and standard deviation) for all datasets being compared. Furthermore, optical depths should be given (either in the text or, preferably, in the figure captions or legends) for all profiles plotted in figure 5 and figures 10–13.

It is also my view that the authors have not responded sufficiently to several of the comments made by the original referees. Below I have listed the original referee comments together with the authors' responses and my criticisms of those responses. I hope the authors will revisit their original responses, and consider adding the additional requested by all referees.

In addition to this review, I have attached an annotated version of the manuscript that contains a number of questions and suggestions. I hope to see responses to these issues reflected in the published version of this paper.

**Referee 1:**

Comments: Specific comments: Section 2, can you briefly describe the AOD measurement uncertainty of these instrument?

*Response: This is a great question. Most validation and uncertainties analysis efforts of satellite AOD retrievals are focus on visible channels. To our knowledge, uncertainties in AOD retrieval at 1064 nm, both from passive and active sensors, are less studied. Just as suggested from the comments from Mark Vaughan and Stuart Young (Short comment for this paper), this paper might be among the first to go deep into AOD retrievals at 1064 nm channel. We were not able to find*

*papers to address uncertainties in AOD retrievals at 1064 nm, although there are papers that do show comparisons between CALIOP and AERONET AOD at 1064 nm (Omar et al., 2013).*

*Omar, A. H., D. M. Winker, J. L. Tackett, D. M. Giles, J. Kar, Z. Liu, M. A. Vaughan, K. A. Powell, and C. R. Trepte (2013), CALIOP and AERONET aerosol optical depth comparisons: One size fits none, J. Geophys. Res.Atmos.,118, 4748–4766, doi:10.1002/jgrd.50330.*

*We have added the following discussion in the text: "Note that most evaluation efforts for passive- and active-based AOD retrievals are focused on the visible spectrum and the performance of AOD retrievals at the 1064 nm channel is less explored. "*

I don't think this response adequately addresses reviewer's request. Estimating measurement uncertainties in not synonymous with validation; ideally, the former would always precede the latter. MODIS AOD uncertainties are explored in numerous papers (e.g., Tanré et al., 1997; Levy et al., 2003; Remer et al., 2005; and many others), and the same is true for AERONET aerosol retrievals (e.g., Holben et al., 1998; Dubovik et al., 2000; Sinyuk et al., 2012; and many others). Uncertainties in extinction and optical depth estimates retrieved using elastic backscatter lidars have a long history in the literature (e.g., Russell et al., 1997; Bissonnette, 1986; Jinhuan, 1988; Young, 1995; Del Guasta, 1998). In particular, the retrieval uncertainties for CALIOP are given in excruciating detail in Young et al., 2013. I'm not aware of any publication that specifically examines CATS extinction uncertainties. However, since CALIOP and CATS are both elastic backscatter lidars that use similar retrieval algorithms, I suspect that the material in Young et al., 2013 could easily be adapted to provide first-order estimates for the CATS uncertainties.

I suspect the authors could make a useful response to this referee's request in just 3 or 4 summary sentences.

(Full disclosure: I am a back-of-the-pack coauthor on Rebecca Pauly's CATS calibration paper.)

In my opinion, this comment should be fully addressed in this paper, and not postponed to some future paper. There are several reasons why "the AOD measured by CATS [might be] less than all other instruments", and the authors should make a good faith attempt to enumerate and discuss at least the most obvious of those reasons.

**Referee 2:**

Comments: (5) The aerosol extinction at 1064 nm may not be as sensitive to the fine mode aerosols (such as smoke and urban pollutant aerosols) compared to the coarse mode aerosols (such as dust). The authors probably should add a few sentences to address this

*Response: Great point. We have added the following discussions to address this issue. "Still, readers  should be aware that AOD retrievals at the 1064 nm are less sensitive to fine mode aerosols such as smoke and pollutant aerosols, compared to coarse mode aerosols such as dust aerosols. Thus, an investigation of diurnal variations of aerosol properties at the visible channel may be also needed for a future study."*

A reference for the first statement would make a very nice addition to the paper.

**Short Comments by Mark Vaughan and Stuart Young**

Comment: While the main body of the text emphasizes the correlations between the CATS retrievals and the other data sets (e.g., lines 186–208), the authors do not provide any quantitative statements about the magnitudes of the CATS AODs or the differences between the different AOD estimates. Given that this paper is (to our knowledge) the first ever in-depth look at 1064 nm AOD, tables showing global and regional mean values and quantifying the CATS AOD estimates relative to the other sensors would add significantly to the value delivered by this paper. Profiles of the relative CATS-CALIOP extinction coefficient differences (i.e., (CATS(z) - CALIOP(z)) / CALIOP(z)) would be especially interesting.

*Response: We have added a table to include regional and global means. Still, we documented that the differences may also be introduced by sampling differences of the sensors.*

*"Table 2. CALIOP and CATS mean aerosol optical depth for regions as highlighted in Figure 6 and globally between +/- 52° latitude."*

| Region | Latitude | Longitude | Mean CATS AOD (DJFMAM/JJASON) | Mean CALIOP AOD (DJFMAM/JJASON) |
|---|---|---|---|---|
| Global | 52°S-52°N | 180°W-180°E | 0.09/0.10 | 0.09/0.09 |
| India | 7.5°N - 32.5°N | 65°E - 85°E | 0.22/0.26 | 0.22 /0.28 |
| Africa - North | 2.5°N - 22.5°N | 35°W - 20°E | 0.26/0.23 | 0.30 /0.25 |
| Africa - South | 17.5°S - 2.5°N | 0° - 30°E | 0.14/0.22 | 0.15 /0.13 |
| Middle East | 12.5°N - 27.5°N | 35°E - 50°E | 0.22/0.33 | 0. 26/0.35 |
| China | 27.5°N - 37.5°N | 110°E - 120°E | 0.19/0.18 | 0.21 /0.16 |

I strongly suggest adding standard deviations to this table; the observed variability of the AODs provides a critically important point of comparison between the two sets of retrievals. I also suggest adding a table comparing CATS means and standard deviations to the AERONET and MODIS means and standard deviations.

Comment: In section 3.1.1., CATS observations are compared with other observations made within ±30 mins and ±0.4 degrees. For aerosols, this is probably not too much of a problem a lot of the time, but we have seen numerous cases where there can be large differences in the scenes being observed (e.g., see Omar et al., 2013: "In 45% of the coincident instances CALIOP and AERONET do not agree on the cloudiness of the scenes."). For AERONET, the comparisons may be improved by imposing another criterion, i.e., that the AERONET AODs made at the closest times preceding and following the CATS observations not vary by more than x%. A similar filter for potential spatial differences could include wind speed and direction (e.g., Lopes et al., 2013) and the spatial separations of the AERONET sites and the CATS observations. (This is likely to be quite a bit messier.)

*Response: We have included the references as suggested and reminded readers that the collocation criteria may have impacts to the results due to the spatial and temporal sampling methods chosen.*

*"Note that as suggested by Omar et al., 2013, the choices of spatial and temporal collocation windows have an effect on collocation results. However, we consider this as a topic beyond the scope of this study"*

While I did not expect the authors to do a complete reanalysis of their data, I had hoped to see a bit more in-depth discussion of the uncertainties inherent in this kind of simple temporal and spatial matching technique and some discussion, perhaps, on how these might be mitigated. For example, the authors use version 3 level 2 AERONET data in their study, whereas the Omar et al., 2013 analysis used version 2 level 2 AERONET data. Are there improvements between versions 2 and 3 that might reduce the differences in the cloudiness inferred by AERONET versus the cloudiness observed by coincident space-based lidar measurements?

Comment: Furthermore, given the lower CATS AODs shown in Figure 2, it's surprising to see that the CATS extinctions coefficients shown in Figure 5 are typically larger than CALIOP at all altitudes, and that the closest agreement is over land (where CATS slightly underestimates

CALIOP at lower altitudes). Again, some discussion of the possible causes of this paradox would be welcome.

*Response: First, there is a call from the community to avoid using slopes from the regression analysis as they are prone to noisy data, and we are kind of agree with them. Statistically, we expect a high percentage of small AODs versus large AODs. Still, slopes are dominated by high AOD cases, while the averaged profiles may be more dominated by low AOD cases. This could explain the difference.*

This response helped motivate my "primary comment" in the opening paragraphs of this review. Given that slopes (and correlation coefficients) are imperfect metrics, additional statistical parameters should be given to more fully characterize the comparisons between the different datasets.

Comment: The CATS extinction profiles shown in Figures 5 and 10 peak at altitudes some hundreds of meters higher than do CALIOP's, except over land. While CALIOP's profiles show almost no roll off until about the last range bin above the surface, the CATS profiles start dropping off below about 500 m, or at approximately 8 to 9 range bins above the surface. What is happening here? Is CATS altitude registration and/or surface detection the culprit? Or is the cloud filter too aggressive in the boundary layer (i.e., are strongly scattering aerosols being misclassified as clouds)? Irrespective of the underlying cause(s), is this behavior a major source of AOD differences between CATS and CALIOP?

*Response: The 2 biggest issues in the CATS V2-01 data were the daytime calibration and the daytime cloud-aerosol discrimination. A CATS paper in preparation (Yorks et al., 2019) has included details about the cloud-aerosol discrimination issues, while Rebecca Pauly's 1064 nm calibration paper has a lot of details about the new daytime calibration. We have checked this issue by reprocessing the analysis using 3 months of V3 data and we found an improvement in agreement for AOD, but with some differences still evident in the vertical profiles.*

While this is a helpful explanation, I do not see where it appears in the revised paper. Given that the CATS V3 data is now publicly available, I think it's essential to include some information that relates these findings to the currently available CATS data.

Comment: The seasonal maps (Figure 6) show that the CALIOP AODs exceed those of CATS over the Arabian Peninsula, and to a smaller degree over the African region bordering the Gulf of Guinea. Can this also be explained by differences in lidar ratio selection, or are there other factors at work?

*Response: We suspect the difference in retrieval method as mentioned above may contribute. Also, CALIOP provides early morning and afternoon overpasses while CATS can observe at near all solar hours, the differences may also be associated with these sampling differences.*

Again, I do not see where this helpful explanation appears in the revised manuscript.

**Specific Comments**

Comment: page 8, line 163: logarithmic interpolation, correct? Also, please state the actual value of the Ångström exponent given by Shi et al.

*Response: Yes. The Angstrom exponent value is computed for each AOD retrieval. We have revised the discussion to avoid confusion. "Here we assume the angstrom exponent value, computed using*

*instantaneous AOD retrievals at the 860 and 1240 nm, remains the same for the 860 to 1064 nm wavelength range, similar to what has been suggested by Shi et al., (2011; 2013)."*

To repeat an earlier comment, please provide a representative range (e.g., mean and standard deviation or some other common statistical description) of the Ångström exponents actually used. Don't leave your readers guessing and/or wondering about what values you used in deriving your 1064 nm AOD estimates.

Comment: page 8, line 170: while "AERONET data are considered as the ground truth for evaluating CATS retrievals", it should be noted that there are very few AERONET sites in remote oceans. Do MODIS retrievals substitute as the gold standard in these places?

*Response: Even though a better performance can be expected from MODIS aerosol retrievals over ocean versus over land, we still think that only AERONET data should be used for ground truth, as instantaneous retrievals from passive sensors suffer from various issues such as cloud contamination.*

This is not a very useful response, mostly because AERONET, like MODIS, is a passive sensor and thus also suffers from "various issues such as cloud contamination" (e.g., Chew et al., 2011 and Huang et al., 2011).

Taking the authors' response at face value, the number of opportunities for ground truth over ocean must be vanishingly small relative to the number of over-ocean measurements being evaluated.

01 data, which have been improved for CATS V3-00 data. A brief analysis using 3 months of

CATS V3-00 data showed improvement in agreement for AOD, but some differences were still evident in the extinction vertical profiles. These remaining differences, as well as the differences observed in nighttime only profiles (Figure 5c) are likely 
[revised manuscript text omitted]

---

## Author Response (AR2)

**Short Comments by Mark Vaughan**

**Comments:**
This paper compares the spatial and temporal distributions of the aerosol optical depths retrieved at 1064 nm by the CATS lidar aboard the International Space Station to the optical depths measured by AERONET (at 1020 nm) and the optical depths retrieved by MODIS and CALIOP (at 1064 nm).

This is the second version of this manuscript that I have read, but the first for which I've been asked to provide an invited (as opposed to contributed) review. My primary comment about this second version is that the authors' do not provide enough information for readers to confidently assess the quality of the CATS AOD estimates relative to those provided by the other sensors. In particular, relying on correlation coefficients alone to characterize the comparisons is insufficient. Consider the two series defined by y2=2x and y4 = 4 x. While y 2 and y4 are perfectly correlated – i.e., they have a correlation coefficient of 1 – in the mean, y4 is twice as large as y2. So in addition to the correlation coefficients they already provide, the authors should also provide means and standard deviations for each of the datasets being compared. While Table 2 is a fine start, more is needed. In a similar vein, regarding figures 1 through 3, black-on-black overplotting of data points in high data density regions reduces the information content of the figures. So, in addition to the figures, the authors should also cite the descriptive statistics (e.g., min, max, median, mean, and standard deviation) for all datasets being compared. Furthermore, optical depths should be given (either in the text or, preferably, in the figure captions or legends) for all profiles plotted in figure 5 and figures 10–13. It is also my view that the authors have not responded sufficiently to several of the comments made by the original referees. Below I have listed the original referee comments together with the authors' responses and my criticisms of those responses. I hope the authors will revisit their original responses, and consider adding the additional requested by all referees.

In addition to this review, I have attached an annotated version of the manuscript that contains a number of questions and suggestions. I hope to see responses to these issues reflected in the published version of this paper.

*Response: We really appreciate Mark Vaughan's valuable comments and made significant changes to this paper. We took the effort and have regenerated all the figures and tables using the newest version of CATS data (V3_00). We find significant differences between raw V2-01 and V3 CATS aerosol products with outliers are significantly reduced in the raw V3 product. However, after the QA steps implemented in the study we found only marginal differences between V2-01 and V3 CATS aerosol products for this study. Most of the conclusions from this study remain largely unchanged after switching to the V3 CATS aerosol data. One plot (similar to Figure 1) is also included in the Appendix using V2-01 CATS aerosol data for a comparison purpose.*

*We have added density plots and included more descriptive statistical analyses (e.g. new tables 1 and 2) as suggested by Dr. Mark Vaughan. We have also included AODs in the figure captions for Figures 5, 10-11 as suggested. AODs for Figures 12 and 13 are included in the Table 3.*

*In addition, we have responded to detailed comments from the reviewer as below.*

General Remarks

**Comments:** "Comments: Specific comments: Section 2, can you briefly describe the AOD measurement uncertainty of these instrument?
Response: This is a great question. Most validation and uncertainties analysis efforts of satellite AOD retrievals are focus on visible channels. To our knowledge, uncertainties in AOD retrieval at 1064 nm, both from passive and active sensors, are less studied. Just as suggested from the comments from Mark Vaughan and Stuart Young (Short comment for this paper), this paper might be among the first to go deep into AOD retrievals at 1064 nm channel. We were not able to find papers to address uncertainties in AOD retrievals at 1064 nm, although there are papers that do show comparisons between CALIOP and AERONET AOD at 1064 nm (Omar et al., 2013). Omar, A. H., D. M. Winker, J. L. Tackett, D. M. Giles, J. Kar, Z. Liu, M. A. Vaughan, K. A. Powell, and C. R. Trepte (2013), CALIOP and AERONET aerosol optical depth comparisons: One size fits none, J. Geophys. Res.Atmos.,118, 4748–4766, doi:10.1002/jgrd.50330. We have added the following discussion in the text: "Note that most evaluation efforts for passive- and active-based AOD retrievals are focused on the visible spectrum and the performance of AOD retrievals at the 1064 nm channel is less explored. "

I don't think this response adequately addresses reviewer's request. Estimating measurement uncertainties in not synonymous with validation; ideally, the former would always precede the latter. MODIS AOD uncertainties are explored in numerous papers (e.g., Tanré et al., 1997; Levy et al., 2003; Remer et al., 2005; and many others), and the same is true for AERONET aerosol retrievals (e.g., Holben et al., 1998; Dubovik et al., 2000; Sinyuk et al., 2012; and many others). Uncertainties in extinction and optical depth estimates retrieved using elastic backscatter lidars have a long history in the literature (e.g., Russell et al., 1997; Bissonnette, 1986; Jinhuan, 1988; Young, 1995; Del Guasta, 1998). In particular, the retrieval uncertainties for CALIOP are given in excruciating detail in Young et al., 2013. I'm not aware of any publication that specifically examines CATS extinction uncertainties. However, since CALIOP and CATS are both elastic backscatter lidars that use similar retrieval algorithms, I suspect that the material in Young et al., 2013 could easily be adapted to provide first-order estimates for the CATS uncertainties. I suspect the authors could make a useful response to this referee's request in just 3 or 4 summary sentences.
References
Bissonnette, L. R., 1986: Sensitivity analysis of lidar inversion algorithms, Appl. Opt., 25, 2122–2125, doi:10.1364/AO.25.002122.
Del Guasta, M., 1998: Errors in the retrieval of thin-cloud optical parameters obtained with a two-boundary algorithm, Appl. Opt., 37, 5522–5540, doi:10.1364/AO.37.005522.
Dubovik, O., A. Smirnov, B. N. Holben, M. D. King, Y. J. Kaufman, T. F. Eck, and I. Slutsker, 2000: Accuracy assessments of aerosol optical properties retrieved from Aerosol Robotic Network (AERONET) Sun and sky radiance measurements, J. Geophys. Res., 105, 9791– 9806, doi:10.1029/2000JD900040.

Holben, B. N., T. F. Eck, I. Slutsker, D. Tanre, J.P. Buis, A. Setzer, E. Vermote, J. A. Reagan, Y. J. Kaufman, T. Nakajima, F. Lavenu, I. Jankowiak and A. Smirnov, 1998: AERONET — A federated instrument network and data archive for aerosol characterization, Rem. Sens. Env., 66, 1-16, doi:10.1016/S0034-4257(98)00031-5.

Jinhuan, Q., 1988: Sensitivity of lidar equation solution to boundary values and determination of the values, Adv. Atmos. Sci., 5, 229–241, doi:10.1007/BF02656784.

Levy, R. C., L. A. Remer, D. Tanré, Y. J. Kaufman, C. Ichoku, B. N. Holben, J. M. Livingston, P. B. Russell and H. Maring, 2003: Evaluation of the Moderate-Resolution Imaging Spectroradiometer (MODIS) retrievals of dust aerosol over the ocean during PRIDE, J. Geophys. Res., 108, 8594, doi:10.1029/2002JD002460.

Remer, L. A., Y. J. Kaufman, D. Tanre, S. Mattoo, D. A. Chu, Martins, J. V., Li, R. R., Ichoku, C., Levy, R. C., R. G. Kleidman, T. F. Eck, E. Vermote, and B. N. Holben , 2005: The MODIS aerosol algorithm, products, and validation, J. Atmos. Sci., 62, 947–973, doi:10.1175/JAS3385.1.

Russell, P. B., T. J. Swissler, and M. P. McCormick, 1979: Methodology for error analysis and simulation of lidar aerosol measurements, Appl. Opt., 22, 3783–3797, doi:10.1364/AO.18.003783.

Sinyuk, A., B. N. Holben, A. Smirnov, T. F. Eck, I. Slutsker, J. S. Schafer, D. M. Giles and M. Sorokin, 2012: Assessment of error in aerosol optical depth measured by AERONET due to aerosol forward scattering, Geophys. Res. Lett., 39, L23806, doi:10.1029/2012GL053894.

Tanré, D., Y. J. Kaufman, M. Herman and S. Mattoo, 1997: Remote sensing of aerosol properties over oceans using the MODIS/EOS spectral radiances, J. Geophys. Res., 102 (D14), 16971–16988, doi:10.1029/96JD03437.

Young, S. A., 1995: Analysis of lidar backscatter profiles in optically thin clouds, Appl. Opt., 34, 7019-7031, doi:10.1364/AO.34.007019.

Young, S. A., M. A. Vaughan, R. E. Kuehn, and D. M. Winker, 2013: The Retrieval of Profiles of Particulate Extinction from Cloud-Aerosol Lidar Infrared Pathfinder Satellite Observations (CALIPSO) Data: Uncertainty and Error Sensitivity Analyses, J. Atmos. Oceanic Technol., 30, 395-428, doi:10.1175/JTECH-D-12-00046.1.

**Response:** *We have included discussions of uncertainties in various places relating to different data products.*

*"AERONET does not have specific guidance on error in the 1020 nm channel, as it is known to have some thermal sensitivities. However, they do report significantly more confidence in version 3 of the data, which has temperature correction (Giles et al., 2018). Error models are ongoing, and for this study we assume double the RMSE, or +/-0.03."*

*"While the uncertainties in MODIS infrared (e.g. 1240 nm) retrievals are less explored, the reported over ocean MODIS DT AOD retrievals are $(+(0.04 + 0.1*AOD), -(0.02 + 0.1*AOD))$ for the green channel (levy et al., 2013). "*

*"The uncertainties in retrieved aerosol extinction, as suggested by Young et al., (2013), is around 0.05–0.5 $km^{-1}$ for the 532 nm channel. Validated against AERONET data, Omar et al., (2013) suggested that 74% and 81% of the CALIOP AOD retrievals are fall within the expected*

*uncertainties (0.05+0.4\*AOD) as suggested by Winker et al., (2009) for the 1064nm channel, for all sky and clear sky conditions respectively."*

*"Although the uncertainties in CATS aerosol retrievals have not yet been documented for the CATS V3-00 extinction and AOD products, much like CALIOP, uncertainties in the calibration and assumed lidar ratios are the primary contributors to the extinction and AOD uncertainties. The uncertainties in the CATS 1064 nm attenuated total backscatter (ATB) is on the order of 7-10% for nighttime and is around 20% for daytime (Pauly et al., 2019), while the uncertainties in the assumed 1064 nm lidar ratios for CATS are 30%. Thus, the CATS 1064 nm extinction (40-70%) and AOD (30-50%) uncertainties are very similar to the corresponding CALIOP's 1064 nm uncertainties."*

**Comment:** Comments: P8, L163, can you describe what constant value of that Angstrom exponent is used here without letting readers to look for that in Shi et al. paper?
Response: We apologize for the confusion. The Angstrom exponent values are computed using instantaneous retrievals. We have revised the text to avoid confusion.
"Here we assume the angstrom exponent value, computed using instantaneous AOD retrievals at the 860 and 1240 nm, remains the same for the 860 to 1064 nm wavelength range, similar to what has been suggested by Shi et al., (2011; 2013)."
Please provide a representative range (e.g., mean and standard deviation or some other common statistical description) of the Ångström exponents actually used.

*Response:*
*Mean and standard deviation were added to the text as suggested.*
*"Mean and standard deviation of Ångström exponents using this method were 0.69 and 0.55, respectively."*

**Comment:** *Comments: Can you provide an explanation on why the AOD measured by CATS less than all other instruments suggested by Figure 1, 2, and 3?*
*Response: We assume that the reviewer is referring to the slope of the regressions in Figures 1-3. Slopes in linear regressions can often be biased by outliers. In Figure 6, which are spatial plots of AODs from CALIOP and CATS, differences are less noticeable for the DJFMAM season. For the JJASON season, CATS AODs are lower at certain regions (Middle East, India, and North Africa) and higher over other regions (South Africa). The cause of those discrepancies, however, is unclear to us. To really explore the issue, it deserves a paper of its own. Thus, we leave this topic to a future paper*
I'm quite perplexed by this response. First, if the slopes are not trustworthy indicators of the correlation between the two data sets, why report them at all? Or, if the authors are concerned that "slopes in linear regressions can often be biased by outliers", why didn't they remove any obvious outliers before plotting their data and computing and reporting the values of the slopes? Second, and perhaps more important, there's at least one plausible and obvious answer to the reviewer's question. According to Rebecca Pauly's CATS calibration paper, the CATS attenuated backscatter coefficients are biased low by ~19% relative both to ground-based Polly measurements and to CALIOP measurements. This low bias in the attenuated backscatter coefficients will invariably lead to low biases in the retrieved optical depths (see the section on calibration and renormalization errors in Young et al., 2013).
(Full disclosure: I am a back-of-the-pack coauthor on Rebecca Pauly's CATS calibration paper.)
In my opinion, this comment should be fully addressed in this paper, and not postponed to some future paper. There are several reasons why "the AOD measured by CATS [might be] less than all other instruments", and the authors should make a good faith attempt to enumerate and discuss at least the most obvious of those reasons.

*Response: Note that based on the slopes of the regression lines shown in Figures 1-3, AODs retrieved by CATS are less than AERONET, CALIOP and DT Aqua MODIS AOD retrievals. As shown in Table 1, however, for the one-to-one collocated datasets, mean CATS AODs (1064 nm) are ~10% higher than AERONET AODs (1020 nm). The CATS AODs are ~3% higher than CALIOP AOD (1064 nm) and are ~5-10% higher than DT MODIS AODs. One possible explanation for this discrepancy is because mean AODs are dominated by low AOD cases and the slopes of the regression relationships are strongly affected by a few high AOD cases. Thus, it is likely that CATS AODs are overestimated at the low AOD ranges and are underestimated at the high AOD ranges. We have included the following discussions in the text.*

*"Note that based on the slopes of the regression lines shown in Figures 1-3, AODs retrieved by CATS are less than AERONET, CALIOP and DT Aqua MODIS AOD retrievals. As shown in Table 1, however, for the one-to-one collocated datasets, mean CATS AODs (1064 nm) are ~10% higher than AERONET AODs (1020 nm). The CATS AODs are ~3% higher than CALIOP AOD (1064 nm) and are ~5-10% higher than DT MODIS AODs. One possible explanation for this discrepancy is because mean AODs are dominated by low AOD cases and the slopes of the regression relationships are strongly affected by a few high AOD cases. Thus, it is likely that CATS AODs are overestimated at the low AOD ranges and are underestimated at the high AOD ranges."*

**Comment:** Comments: (5) The aerosol extinction at 1064 nm may not be as sensitive to the fine mode aerosols (such as smoke and urban pollutant aerosols) compared to the coarse mode aerosols (such as dust). The authors probably should add a few sentences to address this Response: Great point. We have added the following discussions to address this issue. "Still, readers shall should be aware that AOD retrievals at the 1064 nm are less sensitive to fine mode aerosols such as smoke and pollutant aerosols, compared to coarse mode aerosols such as dust aerosols. Thus, an investigation of diurnal variations of aerosol properties at the visible channel may be also needed for a future study."
A reference for the first statement would make a very nice addition to the paper.

*Response: We have added a reference (e.g. Dubovik et al., 2000)*

**Comment:** Short Comments by Mark Vaughan and Stuart Young (table 2)
I strongly suggest adding standard deviations to this table; the observed variability of the AODs provides a critically important point of comparison between the two sets of retrievals. I also suggest adding a table comparing CATS means and standard deviations to the AERONET and MODIS means and standard deviations.

*Response: We have added standard deviations to Table 2 (now new table 3). We have also added new tables 1 and 2 as suggested.*

**Comment:** While I did not expect the authors to do a complete reanalysis of their data, I had hoped to see a bit more in-depth discussion of the uncertainties inherent in this kind of simple temporal and spatial matching technique and some discussion, perhaps, on how these might be mitigated. For example, the authors use version 3 level 2 AERONET data in their study, whereas the Omar et al., 2013 analysis used version 2 level 2 AERONET data. Are there improvements between versions 2 and 3 that might reduce the differences in the cloudiness inferred by AERONET versus the cloudiness observed by coincident space-based lidar measurements?

*Response: We have included a detailed study of the comparisons between CATS AODs and AODs from other instruments (CALIOP, AERONET and MODIS). This include varying of both spatial and temporal collocation windows (new Table 2). Only marginal differences, however, are found from this exercise. We have included in the following discussions.*

*"Also note that as suggested by Omar et al., (2013), the choices of spatial and temporal collocation windows have an effect on collocation results. Thus, we repeated the exercises in Figures 1-3 by doubling the spatial and temporal collocation windows as well as reducing the collocation windows by half. The descriptive statistics of this sensitivity study is included in Table 2. While the number of collocated data pairs are drastically affected by the spatial and temporal collocation window sizes, less significant changes, however, are found in descriptive statistics such as mean, median, and standard deviations of AODs, as well as slopes and correlation values. The slope of DT Aqua MODIS and CATS AODs, however, seems sensitive to changes in collocation methods. Changes in slope of 0.61 to 0.78 are found for the change of temporal collocation window from 15 minutes to 60 minutes with a fixed spatial collocation window of $0.4°$ Latitude/Longitude. "*

*One of the design purposes of the V3 AERONET data is to reduce thin cirrus cloud contamination. We have also included the following discussions:*

*"Note that Version 3 AERONET data are designed to reduce thin cirrus cloud contamination as well as rescue heave aerosol scenes that were misclassified as clouds in previous versions (e.g. Gail et al., 2019)."*

**Comment:** Comment: Furthermore, given the lower CATS AODs shown in Figure 2, it's surprising to see that the CATS extinctions coefficients shown in Figure 5 are typically larger than CALIOP at all altitudes, and that the closest agreement is over land (where CATS slightly underestimates CALIOP at lower altitudes). Again, some discussion of the possible causes of this paradox would be welcome.
Response: First, there is a call from the community to avoid using slopes from the regression analysis as they are prone to noisy data, and we are kind of agree with them. Statistically, we expect a high percentage of small AODs versus large AODs. Still, slopes are dominated by high AOD cases, while the averaged profiles may be more dominated by low AOD cases. This could explain the difference.

This response helped motivate my "primary comment" in the opening paragraphs of this review. Given that slopes (and correlation coefficients) are imperfect metrics, additional statistical parameters should be given to more fully characterize the comparisons between the different datasets.

*Response: We have included detailed descriptive statistics for Figures 1-3 as suggested. Interestingly, the mean CATS AOD is ~3% higher than the mean CALIOP AOD for the collocated CALIOP and CATS data (see prior response to this). Thus, it is not surprising that "CATS extinction coefficients shown in Figure 4 are typically larger than CALIOP at all altitudes."*

**Comment:** Comment: The CATS extinction profiles shown in Figures 5 and 10 peak at altitudes some hundreds of meters higher than do CALIOP's, except over land. While CALIOP's profiles show almost no roll off until about the last range bin above the surface, the CATS profiles start dropping off below about 500 m, or at approximately 8 to 9 range bins above the surface. What is happening here? Is CATS altitude registration and/or surface detection the culprit? Or is the cloud filter too aggressive in the boundary layer (i.e., are strongly scattering aerosols being misclassified as clouds)? Irrespective of the underlying cause(s), is this behavior a major source of AOD differences between CATS and CALIOP?
Response: The 2 biggest issues in the CATS V2-01 data were the daytime calibration and the daytime cloud-aerosol discrimination. A CATS paper in preparation (Yorks et al., 2019) has included details about the cloud-aerosol discrimination issues, while Rebecca Pauly's 1064 nm calibration paper has a lot of details about the new daytime calibration. We have checked this issue by reprocessing the analysis using 3 months of V3 data and we found an improvement in agreement for AOD, but with some differences still evident in the vertical profiles.
While this is a helpful explanation, I do not see where it appears in the revised paper. Given that the CATS V3 data is now publicly available, I think it's essential to include some information that relates these findings to the currently available CATS data.

*Response: We took the time and actually reworked on the paper using V3 CATS data. However, the above mentioned discrepancy still exist. We have suspected the following reasons and included them in the paper:*
*"Due to the precessing orbit of the ISS, the CATS sampling is irregular and very different compared to the sun-synchronous orbits of the A-Train sensors. These orbital differences between CATS and CALIOP make comparing the data from these two sensors challenging since they are fundamentally observing different locations of the Earth at different times. Thus, we shouldn't expect the extinction profiles and AOD from these two sensors to completely agree. Additionally, there are other algorithm and instrument differences that can lead to differences in extinction coefficients and AOD. Over land where dust is the dominant aerosol type, differences in lidar ratios between the two retrieval algorithms (CATS uses 40 sr while CALIOP uses 44 sr), can cause CATS extinction coefficients that are up to 10% lower than CALIOP, potentially explaining the higher CALIOP extinction values in Figure 5e. Over ocean, especially during daytime, differences*

*in CATS and CALIOP lidar ratios for marine and smoke aerosols can introduce a difference between CATS and CALIOP extinction coefficients (Figure 5d). These difference in over ocean data (Figure 5d) could also attributed to differences in CATS and CALIOP 1064 nm backscatter calibration.   For example, Pauly et al. (2019) reports that CATS attenuated total backscatter is about 19.7% lower than PollyXT measurements in the free troposphere and 19% lower than CALIOP of opaque cirrus clouds due to calibration uncertainties for both sensors.*

**Comment:** *Comment: The seasonal maps (Figure 6) show that the CALIOP AODs exceed those of CATS over the Arabian Peninsula, and to a smaller degree over the African region bordering the Gulf of Guinea. Can this also be explained by differences in lidar ratio selection, or are there other factors at work?*
*Response: We suspect the difference in retrieval method as mentioned above may contribute. Also, CALIOP provides early morning and afternoon overpasses while CATS can observe at near all solar hours, the differences may also be associated with these sampling differences.*
Again, I do not see where this helpful explanation appears in the revised manuscript.

**Response:** *It is included:*

*"Those discrepancies may result from biases in each product, but it is also possibly due to the differences in satellite overpass times, as CALIOP provides early morning and afternoon over passes, and Aqua MODIS has an over pass time after local noon, while CATS is able to report atmospheric aerosol distributions at multiple times during a day.  "*

**Comment:**  Comment: page 8, line 163: logarithmic interpolation, correct? Also, please state the actual value of the Ångström exponent given by Shi et al.
Response: Yes. The Angstrom exponent value is computed for each AOD retrieval. We have revised the discussion to avoid confusion. "Here we assume the angstrom exponent value, computed using instantaneous AOD retrievals at the 860 and 1240 nm, remains the same for the 860 to 1064 nm wavelength range, similar to what has been suggested by Shi et al., (2011; 2013)."
To repeat an earlier comment, please provide a representative range (e.g., mean and standard deviation or some other common statistical description) of the Ångström exponents actually used.  Don't leave your readers guessing and/or wondering about what values you used in deriving your 1064 nm AOD estimates.

**Response:** *Done (see prior response to this).*

**Comment:**  Comment: page 8, line 170: while "AERONET data are considered as the ground truth for evaluating CATS retrievals", it should be noted that there are very few AERONET sites in remote oceans. Do MODIS retrievals substitute as the gold standard in these places?
Response: Even though a better performance can be expected from MODIS aerosol retrievals over ocean versus over land, we still think that only AERONET data should be used for ground truth, as instantaneous retrievals from passive sensors suffer from various issues such as cloud contamination.

This is not a very useful response, mostly because AERONET, like MODIS, is a passive sensor and thus also suffers from "various issues such as cloud contamination" (e.g., Chew et al., 2011 and Huang et al., 2011). Taking the authors' response at face value, the number of opportunities for ground truth over ocean must be vanishingly small relative to the number of over-ocean measurements being
evaluated.

*Response: We deleted the sentence to avoid confusion.*

**Comment:** Line 193, "doesn't that cover the entire CATS mission lifetime? since a study covering the whole mission lifetime is about as comprehensive as could be, it might be worth pointing this out."

*Response: As suggested. We have added "the entire mission" in the text.*

**Comment:** Line 208, "you could help other folks that might want to use CATS data by saying which of the QA steps were most effective in removing the large AODs"

*Response: In CATS Version 2, by far the most effective step was limiting the 1064 nm Extinction Coefficient to less than 1.25 $km^{-1}$. However, with the improvements in the CATS Version 3 dataset, this is no longer as clear cut. Still, for a 1% reduction in the number of passes, correlation coefficient between CATS and AERONET is improve from 0.51 to 0.65, standard deviation is reduced from 0.15-0.21 to 0.10-0.12, and several outliers of high CATS AOD/low AERONET AOT are removed. Requiring the Extinction QC flag to be equal to 0 and the Extinction Uncertainty to be less than 10 $km^{-1}$ had the largest impacts on reducing the difference in mean and medians of the AERONET and CATS AOD, with a loss of only 4 passes out of 2270 (less than 0.2% reduction in passes) for each threshold.*

*We added in the text, "We also found that requiring the Extinction QC flag to be equal to 0 and the Extinction Uncertainty to be less than 10 $km^{-1}$ had the largest impacts on reducing the difference in mean and medians of the AERONET and CATS AOD. "*

**Comment:** Lines 214-215, "hence the utility of describing which of the QA steps were most effective in winnowing out unsuitable values"

*Response: See prior comment on this.*

**Comment:** Line 220 "if you interpolate/rescale the MODIS data, why don't you also interpolate/rescale the AERONET data?"

**Response:** *AERONET data include observations at 1020 nm, which is close to 1064nm CATS wavelength, and we expect comparable values for AOD for the two wavelengths. The adjacent MODIS retrievals are at 870 and 1240 nm, thus an interpolation is needed as we expect large changes in AOD from 1020 nm to either 870 nm or 1240 nm.*

**Comment:** Line 226, "please clarify; are MODS 1064 nm AOD estimates obtained via linear interpolation between 870 nm and 1240 nm, or by logarithmic interpolation? make this explicit in the text."

**Response:** *We have added "logarithmic" and "again, assuming the aerosol Ångström Exponent value remains unchanged from 0.87 to 1.064 μm as well as from 1.064 μm to 1.24 μm spectral channels" in the text*

**Comment:** Line 239, "more importantly, CALIOP and CATS are both elastic backscatter lidars (though there are significant differences in instrument design)"

**Response:** *We have added the change as well as included the sentence "Note that despite difference in instrumental designs, CALIOP and CATS are both elastic backscatter lidars."*

**Comment:** Line 246, "disagree; providing some assurance and/or assessment of an apples-to-apple comparison is an essential part of the evidence that should be presented"

**Response:** *We have conducted a detailed sensitivity study relating to collocation window sizes (Table 2). Related discussions are also included as mentioned in a previous response.*

**Comment:** Line 256, "CALIOP daytime 1064 nm data is slightly noisier than CALIOP nighttime 1064 nm data. CATS daytime 1064 nm data is substantially noisier than CATS nighttime 1064 nm data."

**Response:** We removed "expected to be" as suggested. We didn't include the suggested comment as we are unaware of a paper that we can cite for this comment.

**Comment:** Line 258, "a useful adjunct to the AOD study would be a companion study that compared mean vertical extents of the aerosols detected by the two instruments. how much of the AOD differences can be attributed to simple layer detection differences? (this can be answered in part by computing the mean extinction coefficients rather than the mean AODs.)"

**Response:** Nice suggestion. Still, the difference in AOD could be attributed to many factors, including cloud screening, uncertainties in various observed and modeled parameters used in the retrieval process including lidar ratio, as well as other factors.  It is a study of its own and thus we didn't explore the issue further.

**Comment:**  Line 262, "clouds are not "accurately detected by the CATS data products".  instead, the CATS data products report the detection results achieved by the CATS retrieval algorithms and software."

*Response: We have rewritten the sentence as "the feature type score can be used for clouds screening throughout the diurnal envelope of solar angles" to avoid confusion.*

**Comment:**  Line 263, "reconsider this statement; CATS day vs. night differences in SNR pretty much guarantee that there will be some day vs. night detection biases."

*Response:   We have rewritten the sentence as "To further evaluate impact of the solar contamination introduced bias in the diurnal analysis in aerosol detection or products, CATS AODs are evaluated as a function of local time."*

**Comment:**  Line 266, "ugly sentence structure; see the comment from the original Vaughan & Young review referencing the Alan Robock article in EOS ("Parentheses are (are not) for references and clarification (saving space)")"

*Response:   We have revised the sentence as suggested.*

**Comment:**  Lines 281-282, "are the differences statistically significance?  please comment."

*Response:   Comparing the mean AOD at local midnight to the mean AOD at local noon by performing a student's t test, the difference is not significant at the 95% confidence level, with a p-value of 0.16.*
*We have added the following discussions.*
*"Still, comparing the mean AOD at local midnight to the mean AOD at local noon by performing a student's t test, the difference is not significant at the 95% confidence level, with a p-value of 0.16. "*

**Comment:**  Line 304, "disagree; the comparison in 5c does not look substantially better than those shown in 5a, 5b, or 5d.  reporting the optical depths calculated from each of these profiles might help clarify."

*Response:   We have rewritten the sentence to "As shown in Figure 5e, a reasonable agreement is found between CATS V3-00 aerosol extinction with CALIOP for over land"*

**Comment:**  Line 307, "why is this relegated to an appendix?

the x-axis limits on this plot should be tightened up considerably.  ±0.05 would be much, much better and much, much more revealing of the differences between the two data sets."

**Response:** *This figure has been added to the main text (Figure 5f), and the axes limited to ±0.05 as suggested.*

**Comment:** Line 308, "undefined acronym"

**Response:** *Since we use V3 CATS data now, the discussion was deleted.*

**Comment:** Line 315, "undefined acronym"

**Response:** *Since we use V3 CATS data, the discussion was removed*

**Comment:** Line 316, "need to modify this sentence; CATS V3-00 data was publicly released 31 October 2018. see https://eosweb.larc.nasa.gov/project/cats/l2_table"

**Response:** *Since we use V3 CATS data, the discussion was removed*

**Comment:** Line 331, "I don't see this paper in the list of references"

**Response:** *Added.*

*Yorks, J. E., P.A. Selmer, E.P. Nowottnick, S.D. Rodier, M.A. Vaughan, N. Dacic, M.J. McGill, and S.P. Palm, CATS Level 2 Vertical Feature Mask Algorithms and Data Products: An Overview and Initial Assessment, Atmos. Meas. Tech. Discuss., in preparation, 2019.*

**Comment:** Line 337, "no. according to the abstract in the Pauly paper, CATS mean attenuated backscatter is ~19.7% lower than PollyXT measurements in the free troposphere and ~19% lower than CALIOP measurements of opaque cirrus clouds. that the differences are essentially the same when comparing to measurements made by two different instruments using two different targets suggests that the CATS calibration is responsible for the discrepancies seen in this example."

**Response:** *Done. We have revised the sentence to "Pauly et al. (2019) reports that CATS attenuated total backscatter is about 19.7% lower than PollyXT measurements in the free troposphere and 19% lower than CALIOP of opaque cirrus clouds due to calibration uncertainties for both sensors."*

**Comment:** Lines 342-343, "what explains the very different shapes of the CATS and CALIOP profiles in the lowest 250 m? the authors should discuss this."

**Response:** *Differences in the lowest 250 m between CATS and CALIOP extinction profiles are due to how the instrument algorithms detect the surface and near-surface aerosols. Both the CATS and CALIOP feature detection algorithms create a gap between the surface and near-surface aerosol base altitude, despite the possible presence of aerosols in this altitude region.*

*CALIOP has an aerosol base extension algorithm that is designed to (1) detect scenarios when aerosols are present in the bins just above the surface and (2) extend the near-surface aerosol layer base down to the surface (Tackett et al., 2018). However, CATS does not use such an algorithm so false regions of "clear-air" exist between the surface and near-surface aerosol layers.*

*Tackett, J. L., Winker, D. M., Getzewich, B. J., Vaughan, M. A., Young, S. A., and Kar, J.: CALIPSO lidar level 3 aerosol profile product: version 3 algorithm design, Atmos. Meas. Tech., 11, 4129-4152, https://doi.org/10.5194/amt-11-4129-2018, 2018.*

*We have added the discussion in the text.*

*"Also, differences in the lowest 250 m between CATS and CALIOP extinction profiles are observable, which are due to how the instrument algorithms detect the surface and near-surface aerosols. Both the CATS and CALIOP feature detection algorithms create a gap between the surface and near-surface aerosol base altitude, despite the possible presence of aerosols in this altitude region. CALIOP has an aerosol base extension algorithm that is designed to (1) detect scenarios when aerosols are present in the bins just above the surface and (2) extend the near-surface aerosol layer base down to the surface (Tackett et al., 2018). However, CATS does not use such an algorithm so false regions of "clear-air" exist between the surface and near-surface aerosol layers."*

**Comment:** Line 348, "did the authors require that there be a minimum number of extinction samples in each range bin? if so, what was that number?"

***Response:*** *We did not require a minimum number for these range bins.*

**Comment:** Line 362, "are both daytime and nighttime retrievals used to construct these figures? please include this information."

***Response:*** *The figures include both daytime and nighttime retrievals for Figure 6-9. This has been added to the section describing the creation of these Figures.*

"*Both daytime and nighttime retrievals are included in this figure, as well as Figures 7-9.*"

**Comment:** Line 381, "in addition to mean values, also report the standard deviations"

***Response:*** *Since we use V3 CATS data now, the discussion was deleted.*

**Comment:** Line 382, "do these maps of MODIS AODs include only those values that are coincident with the CALIOP footprints? i.e., was the MODIS averaging done in such a way that we should expect a near one-to-one match with the CALIPSO AODs?"

**Response:** *All MODIS data that passed QA steps were used in creating Figures 6a and 6b. Thus, we do not expect a one-to-one match with CALIOP AODs.*

*We added "using all available MODIS DT retrievals that passed QA steps as described in Section 2.3"*

**Comment:** Line 384, "patterns"

**Response:** *Changed to "patterns" suggested.*

**Comment:** Line 400, change to "nearly"

**Response:** *Done*

**Comment:** Lines 408-409, "I don't understand how this conclusion can be drawn from the evidence presented. CATS doesn't retrieve estimates of background aerosol AOD, does it?"

*Response: we deleted this sentence*
*"as global means are dominated by background aerosols that have weak diurnal variations in measured absolute AOD values"*

**Comment:** Line 447 "would be helpful if the text used the same time format that's used in the figure legends (e.g., 6:00 PM vs. 1800)"

**Response:** *We have updated the figure legend to use the same time format as the text to be consistent as suggested.*

**Comment:** Line 447 "is this behavior consistent with AOD patterns seen in AERONET measurements?"

**Response:** *Due to spatial and temporal sampling differences, we didn't pursue this question. E.g., AERONET AODs are column integrated values. Thus it is hard to inter-compare AERONET AODs with CATS extinctions below 1 km.*

**Comment:** Line 457-458," something seems to be missing in this sentence; should it say "so that a near one-to-one transformation..."???"

**Response:** *We revised the sentence to:*
*"Note a near 1 to 1 transformation can be achieved between UTC and local solar time."*

**Comment:** Line 460, "should this be "seasonal"?"

**Response:** *Corrected*

**Comment:** Line 465, "unsightly sentence construction again; heed the Robock comment

*Response:* *We revised the sentence to :*
*"Regional-based analyses are also conducted for 4 selected regions for the DJFMAM season and 5 selected regions for the JJASON season"*

**Comment:** Line 484," suggest revising this; "local morning" at 00:00 AM would more commonly be known as "midnight"."

*Response:* *Changed to "midnight"*

**Comment:** Line 714,-716 table 2, "add standard deviations in addition to means"

*Response:* *Done for the new Table 3.*

**Comment:** Figure 1, "black-on-black overplotting of points in high data density regions reduces the information content of the figures. so in addition to the figures, cite the descriptive statistics (e.g., min, max, median, mean, and standard deviation) for all datasets being compared"

*Response:* *Done. Also included descriptive statistics in Tables 1 & 2.*

**Comment:** Figure 2, "lack-on-black overplotting of points in high data density regions reduces the information content of the figures. so in addition to the figures, cite the descriptive statistics (e.g., min, max, median, mean, and standard deviation) for all datasets being compared"

*Response:* *Done. Also included descriptive statistics in Tables 1 & 2.*

**Comment:** Figure 3, "black-on-black overplotting of points in high data density regions reduces the information content of the figures. so in addition to the figures, cite the descriptive statistics (e.g., min, max, median, mean, and standard deviation) for all datasets being compared"

*Response:* *Done. Also included descriptive statistics in Tables 1 & 2.*

**Comment:** Figure 5, "the integrals of each profile would be very helpful in assessing the differences, and should be reported either in the text or in the figure caption"

**Response:**
*We have added the mean AOD values, which are directly derived from and proportional to the integrals of the profiles, to the caption for Figure 5.*

**Comment:** Figure A1, "why is this relegated to an appendix?

the x-axis limits on this plot should be tightened up considerably.  ±0.05 would be much, much better and much, much more revealing of the differences between the two data sets."

**Response:**   *Done (see previous response to this).*

[revised manuscript text omitted]